# SaFARi: State-Space Models for Frame-Agnostic Representation

**Hossein Babaei**                                        *hb26@rice.edu*
*Department of Electrical and Computer Engineering*
*Rice University*

**Mel White**                                        *mel.white@rice.edu*
*Department of Electrical and Computer Engineering*
*Rice University*

**Sina Alemohammad**                                        *sa86@rice.edu*
*Department of Electrical and Computer Engineering*
*Rice University*

**Richard G. Baraniuk**                                        *richb@rice.edu*
*Department of Electrical and Computer Engineering*
*Rice University*

**Reviewed on OpenReview:** *https://openreview.net/forum?id=UAgxU8gBtv&noteId=2849tqhViA*

## Abstract

State-Space Models (SSMs) have re-emerged as a powerful tool for online function approximation, and as the backbone of machine learning models for long-range dependent data. However, to date, only a few polynomial bases have been explored for this purpose, and the state-of-the-art implementations were built upon a few limited options. In this paper, we present a generalized method for building an SSM with any frame or basis. This framework encompasses the approach known as HiPPO, but also permits an infinite diversity of other possible "species" of SSM, paving the way for improved performance of SSM-based machine learning models. We dub this approach SaFARi: SSMs for Frame-Agnostic Representation.

## 1 Introduction

Modeling sequential data is a cornerstone of modern machine learning, with applications spanning natural language processing, speech recognition, video analysis, and beyond (Zubić et al., 2024; Alemohammad et al., 2021; Nguyen et al., 2022). A fundamental challenge in these domains is the efficient representation of long-range dependence in time-series data, where the goal is to capture and preserve the essential features of the input signal necessary for downstream tasks over extended time horizons while maintaining computational tractability (Hochreiter & Schmidhuber, 1997).

Machine learning approaches, such as recurrent neural networks (RNNs), struggle to learn long-range dependencies due to limited memory horizons (Elman, 1990; Hochreiter & Schmidhuber, 1997; Schuster & Paliwal, 1997; Pascanu et al., 2013). During backpropagation, gradients are repeatedly multiplied by the same weight matrix, causing them to either shrink exponentially (vanish) or grow exponentially (explode). Vanishing gradients prevent the network from updating weights effectively, while exploding gradients lead to unstable training. Although variants like LSTMs (Graves & Schmidhuber, 2005) and GRUs (Cho et al., 2014) address some of these limitations, they often require task-specific parameterization, and cannot generalize across different sequence lengths or timescales.

State-space models (SSMs) present a powerful alternative for online representation of sequential data. By design, SSMs enable the online computation of compressive representations, maintaining a constant-size

memory footprint regardless of sequence length. The seminal work of Gu et al. introduced High-Order Polynomial Projection Operators (HiPPO), which leverages orthogonal function bases to enable theoretically grounded, real-time updates of sequence representations. This framework, and its subsequent extensions into learned architectures, have demonstrated remarkable performance in tasks involving long-range dependencies, such as language modeling and signal processing (Gu & Dao, 2023; Gu et al., 2022b; 2023; Gupta et al., 2024; Gu et al., 2022a; Smith et al., 2023; Hasani et al., 2023). By formulating sequence representation as an online function approximation problem, HiPPO provides a unified perspective on memory mechanisms, offering both theoretical guarantees and practical efficiency.

However, despite its successes, the HiPPO framework has been limited to specific families of orthogonal polynomials. While these bases are well-suited for certain applications, they are not universally optimal for all signal classes. Fourier bases, for instance, are optimal for smooth, periodic signals due to their global frequency representation. Polynomial bases, such as orthogonal polynomials (e.g., Legendre or Chebyshev), are particularly effective for approximating smooth functions over compact intervals.

The absence of a more flexible basis selection restricts the adaptability of the HiPPO framework. In this work, we address this restriction by presenting a generalized method for constructing SSMs using any frame or basis of choice, which we term SaFARi (SSMs for Frame-Agnostic Representation). Our approach extends and generalizes the HiPPO framework using a numerical (as opposed to closed-form) method, which enables us to relax the requirements on the basis, such as orthogonality of the components.

Our key contributions are as follows:

- **Generalized SSM construction:** We present SaFARi, a frame-agnostic method for deriving SSMs associated with any basis or frame, generalizing the HiPPO framework to a broader class of function representations.

- **Error Analysis:** We provide a comprehensive discussion of SaFARi's error sources and derive error bounds, offering theoretical insights into its performance and limitations.

This paper is organized as follows. In Section 2, we review the HiPPO framework and its limitations, motivating the need for a generalized approach. Section 3 provides the required mathematical preliminaries. Section 4 introduces our frame-agnostic method for SSM construction, and then Sections 5 and 6 address implementation considerations and strategies for SaFARi, including error analysis and computational efficiency. Section 7 demonstrates empirical validation of the method and the theoretical claims presented in this paper. Finally, Section 8 discusses the broader implications of our work and outlines directions for future research.

## 2 Background

Recent advances in machine learning, computer vision, and LLMs have exploited the ability to collect and contextualize more and more data over longer time frames. Handling such sequential data presents three main challenges: 1) generating a compact and low-dimensional representation of the sequence, 2) effectively preserving information within that representation, and 3) enabling real-time updates for streaming data.

The classic linear method of obtaining the coefficients of a compressed representation of a signal is through a transform (e.g. Fourier) (Oppenheim, 1999; Abbate et al., 2012; Box et al., 2015; Proakis, 2001; Prandoni & Vetterli, 2008). However, a significant limitation of this method is its inefficiency in handling real-time updates. When new data arrives, the representation must be recalculated in its entirety, necessitating the storage of all prior data, which is inefficient in terms of both computation and storage requirements. This limits the horizon of the captured dependencies within the sequence.

Nonlinear models, such as recurrent neural networks (RNNs) and their variants, have been introduced more recently (Elman, 1990; Hochreiter & Schmidhuber, 1997; Cho et al., 2014; Schuster & Paliwal, 1997). Since these learned representations are task-specific, they are not easily utilized for other circumstances or applications. Furthermore, RNNs struggle to capture long-range dependencies due to issues such as vanishing and exploding gradients.

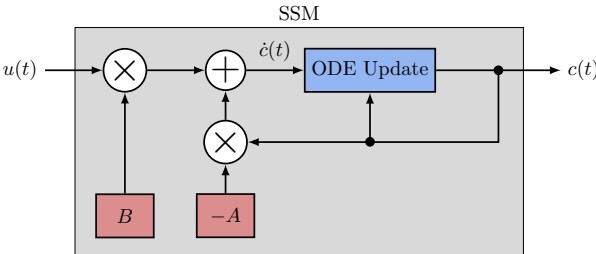

Figure 1: An SSM block-diagram, with the necessary ODE update step included.

## 2.1 State-space models

The state-space representation itself is not new; it was introduced by Kálmán (1960) via the eponymous Kalman Filter. For an input $u(t)$, output $y(t)$, and a state representation called $x(t)$, many systems and their properties can be described and controlled with the following system of ODEs, illustrated in Fig. 1:

$$\dot{x}(t) = Ax(t) + Bu(t)$$
$$y(t) = Cx(t) + Du(t). \tag{1}$$

In many classic applications, we iteratively update the matrices $A$, $B$, $C$, and $D$ to control or predict the output $y$ based on previous values of $u$. For online function approximation, however, we instead *define* the matrices $A$ and $B$ such that they iteratively update a vector of coefficients $c$ over a particular basis. For the moment, we can ignore $C$ and $D$ (or, equivalently, consider $C$ to be an identity matrix and $D = 0$). For stability, $A$ must have only negative eigenvalues, so we explicitly include a negative sign here. $A$ and $B$ may or may not be constant over time, so for completeness, we call these $A(t)$, $B(t)$. Eq. 1 is now

$$\dot{c} = -A(t)c(t) + B(t)u(t). \tag{2}$$

The challenge in the problem of the approximation of online functions is to derive appropriate matrices $A$ and $B$. Legendre Memory Units (LMUs) were proposed for this task by (Voelker et al., 2019), and subsequently expanded upon by (Gu et al., 2020). We note that the term "SSM" in machine learning literature in recent years has become a synecdoche, referring both to the classical $A$, $B$ matrix structure described here, as well as the larger models that utilize these SSMs as a layer, such as S4 Gu et al. (2022b). For the purposes of this paper, we disambiguate these structures, and we will use the term SSM to describe only the former.

## 2.2 HiPPO: High-order Polynomial Projection Operators

The LMU and HiPPO frameworks (Voelker et al., 2019; Gu et al., 2020) enable online function approximation using pre-determined SSM parameters derived from a basis of orthogonal polynomials $\mathcal{G}$. It optimizes $\|u_T - g^{(T)}(t)\|_\mu$ for $g^{(T)}(t) \in \mathcal{G}$, to find a set of coefficients for the orthogonal polynomials at every time $T$, which yields the approximation of the input function over $[0, T]$.

In addition to choosing the set of polynomials, one must also select a measure, $\mu$: the combination of a weighting function and a windowing function. The window indicates which samples are included, and the weighting assigns a weight to each sample. HiPPO (Gu et al., 2020) considered several possible measures, two of which (illustrated in Fig. 2) are:

$$\mu_{tr}(t) = \frac{1}{\theta}\mathbb{1}_{t\in[T-\theta,T]}, \quad \mu_{sc}(t) = \frac{1}{T}\mathbb{1}_{t\in[0,T]}. \tag{3}$$

The uniform translated measure, $\mu_{tr}(t)$, gives equal weight to all samples in the most recent window with a constant length $\theta$, and zero weight to previous samples. The uniform scaled measure, $\mu_{sc}(t)$, gives equal weight to all the times observed since $t = 0$. This can be interpreted as squeezing or stretching the basis or frame to match the current length of the signal at any given time. Thus, the representation produced by this measure becomes less informative about the finer details of the signal as more history is observed, since

the stretching of the basis will gradually increase the lowest observable frequency. This work considers only uniformly weighted measures, for the sake of clarity in our derivations and examples. However, one could implement any desired weighting scheme by applying any function of $T$ in place of $\frac{1}{\theta}$ or $\frac{1}{T}$ in Eq. 3.

For the space of orthogonal polynomials, the HiPPO ODE update (Gu et al., 2020) (see Fig. 1) follows a first-order linear differential equation:

$$\frac{d}{dT}\vec{c}(T) = -A_{(T)}\vec{c}(T) + B_{(T)}u(T) \tag{4}$$

where $u(T)$ is the value of the input signal at the current time $T$, and $\vec{c}(T)$ is the vector containing the representation of the $t \in [0, T]$ part of the input signal. $A_{(T)}$ and $B_{(T)}$ are also a time-varying matrix and vector that can be derived for the particular choice of the measure and orthogonal polynomial.

**Solving the differential equation to update the SSM**: The differential equation (Eq. 4) can be solved incrementally, and there are several methods to do so Butcher (1987). We use the *generalized bilinear transform (GBT)* (Zhang et al., 2007), relying on findings from Gu et al. (2020) that the GBT produces the best numerical error in solving first-order SSMs. The GBT update rule is

$$c(t + \Delta t) = (I + \delta t \alpha A_{(t+\delta t)})^{-1}(I - \delta t(1 - \alpha)A_{(t)})c(t) + (I + \delta t \alpha A_{(t+\delta t)})^{-1}\delta t B(t)u(t), \tag{5}$$

where $0 \le \alpha \le 1$ is a chosen constant. (For $\alpha = 0$, the update rule becomes the forward Euler method.)

**Diagonalizing the transition matrix $A$**: The incremental update given by the GBT requires a matrix inversion and matrix products at each increment. Having a diagonal $A$ significantly reduces the computational cost of these operations. If the measure used for the SSM is such that the eigenvectors of $A(t)$ remain constant (for example, if $A(t)$ is a constant matrix multiplied by an arbitrary function of time), then it is possible to find a change of basis that makes the SSM matrix diagonal. To do this, one finds the eigenvalue decomposition of the matrix $A(t) = V\Lambda(t)V^{-1}$ and re-write the SSM as

$$\frac{\partial}{\partial t}\widetilde{c} = -\Lambda(t)\widetilde{c} + \widetilde{B}u(t), \tag{6}$$

where $\widetilde{c} = V^{-1}c$ and $\widetilde{B} = V^{-1}B(t)$. This means that one can solve Eq. 6 instead of Eq. 4, then find the representation $c$ from $\widetilde{c}$ with a single matrix multiplication.

## 2.3 Limitations of HiPPO

The original HiPPO formulation and a subsequent follow-up (Gu et al., 2023) included a handful of orthogonal polynomial bases with specific measures. There is no method in the literature for arbitrary choice of basis and measure (see Table 1).

---

[1]S4 technically applies an exponential measure in the ODE; however, the $A$ matrix of the SSM is generated based on a uniformly weighted scaled Legendre polynomial basis.

[2]Some implementations in the translated case also apply an exponential decay measure in order to ensure orthogonality.

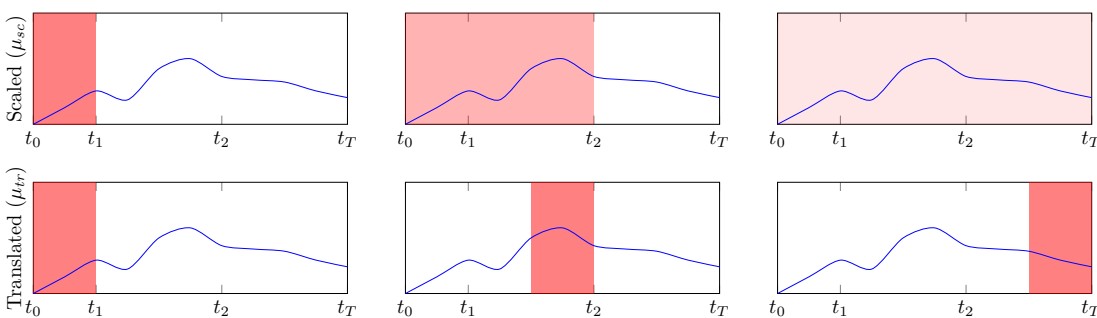

Figure 2: Two different uniform measures (red) applied to a signal (blue). The red shaded area demonstrates how the measure changes as time evolves and more samples of the input are observed.

|  | Measure | Scaled | Translated |
|---|---|---|---|
| **Basis or Frame** | Legendre | Gu et al. (2020),§4 Gu et al. (2023)[1] | Gu et al. (2020),§4 |
|  | Fourier | Gu et al. (2020), §4 | Gu et al. (2020),§4 |
|  | Laguerre | §4 | Gu et al. (2020)[2],§4 |
|  | Chebychev | §4 | Gu et al. (2020)[2]§4 |
|  | Arbitrary | §4 | §4 |

Table 1: An overview of the combinations of frames (or bases) and measures covered in the literature to date. SaFARi fills in missing combinations in this table for the scaled and translated measures in Section 4.

Performance of various bases were shown to be strongly task-dependent. Only one basis-measure combination, the scaled-Legendre (LegS), performed empirically well across most tasks Gu et al. (2020), but it introduced additional challenges as its $A$ matrix is not stably diagonalizable (Gu et al., 2022a). The majority of the follow-up work since HiPPO has abandoned the task of function approximation by SSMs alone. Instead, most research has employed a diagonal approximation of the best-performing extant version (LegS) as an *initialization* for machine learning architectures such as S4 (Gu et al., 2022b; Smith et al., 2023). Still, the HiPPO framework still holds untapped potential for online function approximation and better initializations for learning tasks.

## 3 Mathematical Preliminaries

Prior to introducing SaFARi, we first cover some required theoretical background on the use of frames for function representation and approximation. This section will cover distinctions between approximations performed on the full signal all at once, and approximation performed in a sequential (online) fashion.

### 3.1 Function representation using frames

Given a function $u(t)$ of a time domain $t \in \mathcal{D}_T$, we aim to represent $u(t)$ using a collection of functions over $\mathcal{D}_T$. This representation is performed with respect to a measure that determines the relative importance of the function's value at different times. We formulate the task of function representation using frames via the definitions below.

**Frame:** The sequence $\Phi = \{\phi_n\}_{n \in \Gamma}$ is a frame in the Hilbert space $\mathcal{H}$ if there exist two constants $A > 0$ and $B > 0$ such that for any $f \in \mathcal{H}$:

$$A_{\text{frame}} \|f\|^2 \leq \sum_{n \in \Gamma} |\langle f, \phi_n \rangle|^2 \leq B_{\text{frame}} \|f\|^2 \tag{7}$$

where $\langle \cdot, \cdot \rangle$ is the inner product of the Hilbert space $\mathcal{H}$, and $\Gamma$ denotes the indices of the frame elements in $\Phi$. If $A_{\text{frame}} = B_{\text{frame}}$, then the frame is said to be tight (Mallat, 2008; Christensen, 2003; Gröchenig, 2001).

**Frame operator:** If the sequence $\Phi = \{\phi_n\}_{n \in \Gamma}$ is a frame, then its associated operator $\mathbb{U}_\Phi$ is defined as:

$$\mathbb{U}_\Phi f = \vec{c}, \qquad c_n = \langle f, \phi_n \rangle, \quad \forall n \in \Gamma. \tag{8}$$

It can be shown that the frame condition (Eq. 7) is necessary and sufficient for the frame operator (Eq. 8) to be invertible on its image with a bounded inverse. This makes the frame a complete and stable (though potentially redundant) framework for signal representation.

**Dual frame:** The set $\widetilde{\Phi} = \{\widetilde{\phi}_n\}_{n \in \Gamma}$ is the dual of the frame $\Phi$, $\widetilde{\Phi} = \text{Dual}\{\Phi\}$ if:

$$\widetilde{\phi}_n = (\mathbb{U}_\Phi^* \mathbb{U}_\Phi)^{-1} \phi_n \tag{9}$$

where $\mathbb{U}_\Phi^*$ is the adjoint of the frame operator: $\langle \mathbb{U}_\Phi f, c \rangle = \langle f, \mathbb{U}_\Phi^* c \rangle$.

It can be shown that the composition of the dual frame and the frame is the identity operator in the Hilbert space $\mathcal{H}$: $\mathbb{U}_{\widetilde{\Phi}}^* \mathbb{U}_\Phi f = f$ (Mallat, 2008; Christensen, 2003; Gröchenig, 2001). Thus, we can think of the dual frame as the operator that transforms the signal from frame representation back into the signal space.

**Function approximation using frames:** The compressive power of frame-based function approximation lies in its ability to efficiently represent functions using a relatively small number of frame elements. Different classes of functions exhibit varying effectiveness in capturing the essential features of different signal classes. This efficiency is closely tied to the decay rate of frame coefficients, which can differ significantly between frames for a given input function class. As a result, selecting an appropriate frame is critical for optimal approximations while minimizing computational resources and storage space.

In the task of online function representation, we aim to represent a function $u_T := \{u(t), t \in [0, T]\}$ for any time $T$, using a frame $\Phi$ that has the dual $\widetilde{\Phi}$ and the domain $t \in \mathcal{D}$. To do so, we need an invertible warping function so that the composition of the frame and the warping function $\Phi^{(T)}$ includes the domain $[0, T]$. Without loss of generality, we assume that the frame has the domain $\mathcal{D}_\Phi = [0, 1]$, and use the warping $\phi^{(T)}(t) = \phi\left(\frac{t}{T}\right)$. (If this is not the case, then one can easily find another warping function to warp $\mathcal{D}_\Phi \rightarrow [0, 1]$ and apply it beforehand.) To calculate the representation, the frame operator of $\Phi_T$ acts on $u_T$:

$$\textit{Projection:} \quad c = \Phi^{(T)} u_T, \quad c_n = \langle u_T, \phi_n^{(T)} \rangle = \int_{t=0}^{T} u(t) \overline{\phi_n^{(T)}(t)} \mu(t) dt, \tag{10}$$

where $\overline{\phi}(t)$ is the complex conjugate of $\phi(t)$. Then, the dual frame operator transforms the discrete representation back to the function $u_T$:

$$\textit{Reconstruction:} \quad u_T = \widetilde{\Phi}^{(T)} \Phi^{(T)} u_T, \quad u_T(t) = \sum_{n \in \Gamma} \langle u_T, \phi_n^{(T)} \rangle \widetilde{\phi}_n^{(T)}(t). \tag{11}$$

**Measures:** Throughout this paper, we work with the Hilbert space consisting of all functions over the domain $\mathcal{D}$ with $\mu(t)$ representing a measure as in Eq. 3, and the inner product defined as:

$$\langle u, v \rangle = \int_{t \in \mathcal{D}} u(t) \overline{v(t)} \mu(t) dt. \tag{12}$$

We will consider both scaling and translating windows with uniformly weighted measures as described in Eq. 3 and Fig. 2, and derive the SSMs for each. We only implement uniform weighting schemes in this work for the sake of clarity and generalizability, as the weighting does not impact the SSM's derivation. The windowing function does impact the derivation, however, because it changes the limits of integration, and so we provide frameworks for both scaled (with a changing number of prior inputs) and translated (with a constant number of prior inputs). Given the combination of these generic models, one could create any arbitrary measure by combining the appropriate window with any desired weighting scheme as a function of $t$ in Eq. 12.

## 4 SaFARi

We now introduce a generalized method for online function representation using arbitrary frames. We formulate the update rule for online approximation as a first-order linear differential equation following a state-space model, and provide templates for the uniform translated and scaled measures.

### 4.1 Formulation

For the given frame $\Phi$ (without loss of generality, we assume $\Phi$ has the domain $\mathcal{D} = [0, 1]$), and an input function $u(T)$, the objective is to find the representation of the function using the frame $\Phi$ similar to Eq. 10:

$$\textit{Projection:} \quad c_n(T) = (\mathbb{U}_\Phi u)_n = \int_{t=t_0}^{T} u(t) \overline{\phi_n^{(T)}(t)} \mu(t) dt. \tag{13}$$

The representation vector $\vec{c}(T)$ is a vector with its $n^{\text{th}}$ component defined as above. We next find the derivative of the representation with respect to the current time $T$, and show that it follows a particular SSM for a scaled or translated measure.

## 4.2 Uniform scaled measure: $\mu_{sc}$

When we use the scaled measure ($\mu_{sc}(t)$ in Eq. 3), the representation generated by applying the frame operator to the observed history of the input $u_T(t)$ is

$$c_n(T) = \int_{t=0}^{T} u(t)\overline{\phi_n}\left(\frac{t}{T}\right)\frac{1}{T}dt\,. \tag{14}$$

**Definition 1.** For a given frame $\Phi$ consisting of functions on $\mathcal{D}$, we define the auxiliary derivative of the frame $\Upsilon_\Phi = \{\upsilon_n\}_{n\in\Gamma}$ as a collection having $\upsilon_n = t\frac{\partial}{\partial t}\phi_n(t)$.

The auxiliary derivative of the frame is the result of the operator $t\frac{\partial}{\partial t}$ acting on each individual frame element. Note that the auxiliary derivative of the frame is not necessarily a frame itself and the frame condition in Eq. 7 does not necessarily hold for $\Upsilon_\Phi$.

**Theorem 1.** *For the representation defined in Eq. 14, the partial derivative of $\vec{c}$ with respect to $T$ is*

$$\frac{\partial}{\partial T}\vec{c}(T) = -\frac{1}{T}A\vec{c}(T) + \frac{1}{T}Bu(T), \quad A = I + \mathbb{U}_\Upsilon\mathbb{U}_{\widetilde{\Phi}}^* \tag{15}$$

*where $B$ is the complex conjugate of a vector containing members of the main frame evaluated at $T = 1$ so that $B = \{\overline{\phi}_n(T = 1)\}_{n\in\Gamma}$. The $A$ operator can also be described as a matrix*

$$A_{i,j} = \delta_{i,j} + \int_0^1 \left[t\frac{\partial}{\partial t}\overline{\phi}_i(t)\right]_{t=t'}\widetilde{\phi}_j(t')dt'\,. \tag{16}$$

Proof is provided in the Appendix A.1. We will refer to the SSM with a scaling measure as scaled-SaFARi.

## 4.3 Uniform translated measure: $\mu_{tr}$

When the translated measure is used ($\mu_{tr}(t)$ in Eq. 3), the representation resulting from applying the frame operator to the window of recently observed input $u_T(t)$ is

$$c_n(T) = \int_{t=T-\theta}^{T} u(t)\overline{\phi_n}\left(\frac{t-(T-\theta)}{\theta}\right)\frac{1}{\theta}dt\,. \tag{17}$$

**Definition 2.** For a given frame $\Phi = \{\phi_n\}_{n\in\Gamma}$ consisting of functions of time, we define the time derivative of the frame $\dot{\Phi} = \{\dot{\phi}_n = \frac{\partial}{\partial t}\phi_n(t)\}_{n\in\Gamma}$ as a collection of time derivatives of $\phi_n$ components.

**Definition 3.** For a given frame $\Phi = \{\phi_n\}_{n\in\Gamma}$ consisting of functions of time, the zero-time frame operator is similar to the frame operator but only acts on $t = 0$ instead of the integral over the entire domain:

$$\mathbb{Q}_\Phi f = \vec{x}, \quad x_n = f(t = 0)\overline{\phi}_n(t = 0)\,. \tag{18}$$

**Theorem 2.** *For the representation defined in Eq. 14, the partial derivative of $\vec{c}$ with respect to $T$ is*

$$\frac{\partial}{\partial T}\vec{c}(T) = -\frac{1}{\theta}A\vec{c}(T) + \frac{1}{\theta}Bu(T), \quad A = \mathbb{U}_{\dot{\Phi}}\mathbb{U}_{\widetilde{\Phi}}^* + \mathbb{Q}_\Phi\mathbb{Q}_{\widetilde{\Phi}}^* \tag{19}$$

*where $B$ is the complex conjugate of a vector containing members of the main frame evaluated at $T = 1$ so we have $B = \{\overline{\phi}_n(T = 1)\}_{n\in\Gamma}$. The $A$ operator can also be described using the matrix representation*

$$A_{i,j} = \overline{\phi}_i(0)\widetilde{\phi}_j(0) + \int_0^1 \left[\frac{\partial}{\partial t}\overline{\phi}_i(t)\right]_{t=t'}\widetilde{\phi}_j(t')dt'\,. \tag{20}$$

Proof is provided in Appendix A.2. We will refer to the SSM with a scaling measure as translated-SaFARi.

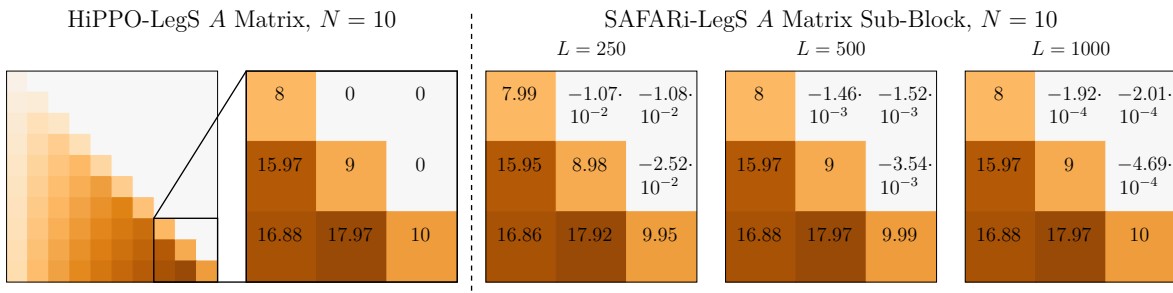

Figure 3: HiPPO provides a closed-form solution for the scaled Legendre (LegS) SSM. SaFARi provides a computed solution, where the accuracy depends on the discretization of the $N \times L$ frame. Larger $L$ gives a finer discretization of the basis vectors and thus a better numerical result.

## 4.4 SaFARi as generalization of HiPPO

HiPPO provides exact, closed-form solutions for $A$ and $B$ for a few specific basis and measure combinations. SaFARi replicates these $A$ and $B$ matrices to within some numerical error caused by discretization of the frame vectors into length $L$. Increasing $L$ will provide matrices that converge toward the closed-form solution (see Fig. 3). When a closed-form solution exists for the desired basis and measure (e.g., HiPPO-LegS for the scaled Legendre basis), then it is preferable to use it Gu et al. (2020; 2023).[1]. SaFARi provides a method for *any* basis/measure where the closed-form solution might not exist. We also show that SaFARi preserves all of HiPPO's robustness to timescale when applied to a general frame in Appendix A.3.

While the numerical method described above could be applied to any differentiable set of vectors, we require that the vectors form a frame. If not, then projecting the input signal onto the given vectors is lossy and not invertible. More precisely, the frame condition (Eq. 7) is necessary and sufficient for the frame operator to be invertible on its image with a bounded inverse. This makes the frame a complete and stable (though potentially redundant) framework for signal representation. Second, and most importantly for this work, if $\phi$ does not meet the frame condition, then Eq. 9 does not hold, and the derivative of representation with respect to time cannot be calculated using only the current hidden state.

## 5 Error analysis

This section describes the computational efficiency and accuracy concerns of SaFARi, including strategies for producing the finite-dimensional approximation of the complete infinite-dimensional SaFARi in Section 5.1. We analyze the errors introduced by these approximations in Section 5.2.

## 5.1 Truncation of frames

Section 4 demonstrates how a particular SSM can be built from an arbitrary frame $\Phi$. Since the input space for the SSM is the class of functions of time, no $\Phi$ with a finite number of elements can meet the frame condition (Eq. 7), since the true representation of the input signal is infinite-dimensional. In practice, the representation reduces to the truncated representation. In this section, we analyze the theoretical implications of truncated representation using SaFARi.

### 5.1.1 Finite-dimensional approximation of SaFARi

In the finite-dimensional case, we will use only $N$ elements of a frame. Partial representation of size $N$ requires that the resulting representation approximates the infinite-dimensional representation. We call the SSM with its $\vec{c}$ having $N$ coefficients SaFARi$^{(N)}$.

---

[1]Note that HiPPO used the convention of absorbing a negative sign from the ODE in Eq. 4 into the $A$ matrix, whereas we do not.

**Definition 4.** A SaFARi$^{(N)}$ sequence is a sequence of the pairs $[A^{(N)}, B^{(N)}]$ where $A^{(N)} \in \mathbb{C}^{N \times N}$ and $B^{(N)} \in \mathbb{C}^N$ such that sequence converges to $[A, B]$ of SaFARi as

$$\lim_{N \to \infty} A^{(N)} = A, \qquad \lim_{N \to \infty} B^{(N)} = B. \tag{21}$$

where convergence for $A$ is the Strong Operator Topology (SOT) convergence, and convergence for $B$ is the vector norm-2 convergence. This convergence means any arbitrary precision can be achieved by selecting the appropriate truncation level. See Appendix A.5 for details.

Definition 4 is not a constructive definition; that is, it does not uniquely determine $[A^{(N)}, B^{(N)}]$. In fact, there may be many such sequences that converge to $[A, B]$. Of course, this does not mean that all such sequences would produce equal representation error. For the following section, we assume the convergence and present and analyze two alternate constructions. See Appendix A.5 for details.

### 5.1.2   Truncation of dual (ToD)

The ToD construct of a SaFARi$^{(N)}$ sequence begins with the infinite-dimensional $A$ and $B$, then truncates to size $N$ as

$$A^{(N)} = A_{[0:N,0:N]}, \quad B^{(N)} = B_{[0:N]}. \tag{22}$$

This construction results in a sequence that approximates the infinite-dimensional SaFARi according to Definition 4. In practice, we find the truncated $A, B$ via $A_{i,j}$ as introduced in Eq. 16 and Eq. 20 for $i, j < N$.

Calculating $A_{i,j}$ requires finding $\widetilde{\Phi}$, the dual of the (infinite-dimensional) frame $\Phi$. For certain families of functions, the dual frame can be found analytically. However, if an analytical dual frame $\widetilde{\Phi}$ is not known, then one must use a numerical approximation of the dual frame. In this case, the construction for $[A^{(N)}, B^{(N)}]$ involves forming the truncated frame for a much bigger size $N_2 \gg N$, then finding $\widetilde{\Phi}^{(N_2)} = \text{Dual}\{\Phi_{[0:N_2]}\}$ numerically as an approximation for the dual frame ($\widetilde{\Phi}^{(N_2)} \approx \widetilde{\Phi}$). Next, we truncate the approximate dual and use its first $N$ elements as an approximation for size $N$ dual frame in Eq. 16 and Eq. 20.

### 5.1.3   Dual of truncation (DoT)

The ToD construction becomes numerically intractable for cases where the dual frame is not analytically known; this motivates the need for an alternate constructor for SaFARi$^{(N)}$. To construct this sequence

1. Truncate the frame at level $N$ and form $\Phi^{(N)} = \{\phi_i\}_{i<N}$.

2. Numerically approximate the dual of the truncated frame $\widetilde{\Phi}^{(N)} = \text{Dual}\{\Phi^{(N)}\}$ using the pseudo-inverse.

3. Use $\Phi^{(N)}$ and $\widetilde{\Phi}^{(N)}$ in Eq. 16 or Eq. 20 to compute $[A^{(N)}, B^{(N)}]$.

The DoT and ToD constructions approximate SaFARi with different rates. In the next sections, we provide a thorough error analysis of SaFARi$^{(N)}$ that enables us to compare the usage of different frames, as well as different construction methods for size $N$ constructs. We then demonstrate that the DoT construction always has a minimum reconstruction error, and is the optimal choice for implementing SaFARi.

### 5.2   Error analysis

Using truncated representations for online function approximation will result in some reconstruction error, regardless of basis. We focus here on errors emanating from truncation of the representation in an SSM, rather than those caused by sampling, which have been extensively studied in the digital signal processing literature (Oppenheim, 1999).

Let $c^{(\infty)}$ denote the infinite-dimensional representation obtained from an SSM without truncating its associated $A, B$ matrices. When the associated $A, B$ matrices for the SSM are truncated to the first N levels, it produces a truncated representation $c^{(N)}$. This truncation causes two distinct types of error, which we outline below.

### 5.2.1 Truncation errors

Truncation errors are due to the fact that the truncated frame of size $N$ cannot represent any part of the signal contained in indices $n > N$. This is not limited to SSMs, but is true for any basis representation of a signal. In SaFARi, truncation errors correspond to discarding the green shaded region in the $A^{(N)}$ illustrated in Fig. 4.

### 5.2.2 Mixing errors

Mixing errors arise from error propagation in the SSM update rule. Specifically, this update involves computing $Ac$ (as in Eq. 15 and Eq. 19), where the matrix $A$ introduces unwanted mixing between the omitted components ($n > N$) and the retained components ($n \leq N$) of the representation. Consequently, errors from the truncated portion of the representation propagate into the non-truncated portion. This is illustrated by the blue shaded regions in Fig. 4. For the operator $A$ and truncation level $N$, the contaminating part of the operator is $A_{i,j} \forall (i \leq N, j > N)$.

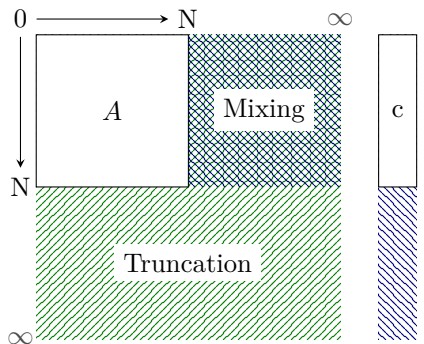

Figure 4: Error types due to frame truncation. Truncation errors arise from energy in coefficients of index $n > N$, while mixing errors result from energy blending during the operation $Ac$.

In the case of a translated measure, this mixing error is exacerbated since each update step requires estimating the initial value in that window (recall Eq. 2). A key insight from our analysis below is that mixing errors have two sources:

1. nonzero components in the upper right quadrant of $A$, and
2. nonzero coefficients in $c$ at indices greater than $N$.

### 5.2.3 Mitigating errors

Truncation errors can never be eliminated, but may be alleviated by using a frame that exhibits a rapid decay in the energy carried by higher-order levels of representation.

To counter mixing errors, we should ensure that values in the upper right quadrant of $A$ are as close to zero as possible, and/or that coefficents of $c$ in the blue region of Fig. 4 are as close to zero as possible. If the matrix $A$ is lower-triangular, then any arbitrary truncation results in the contaminating part of $A$ being zero, which guarantees the second type of error is always zero, regardless of any coefficients in $c$. This is the case for the HiPPO-LegS (scaled Legendre) $A$ matrix, as shown in Fig. 5(a). Indeed, the zero coefficents in the upper right quadrant of $A$ were considered strictly necessary in prior work (Gu et al., 2020; 2023). This restriction explains the continued use of HiPPO-LegS in follow-up works, regardless of whether or not scaled Legendre polynomials are an optimal choice for a given application.

To summarize, there are two primary concerns when finding an appropriate frame for use in an SSM:

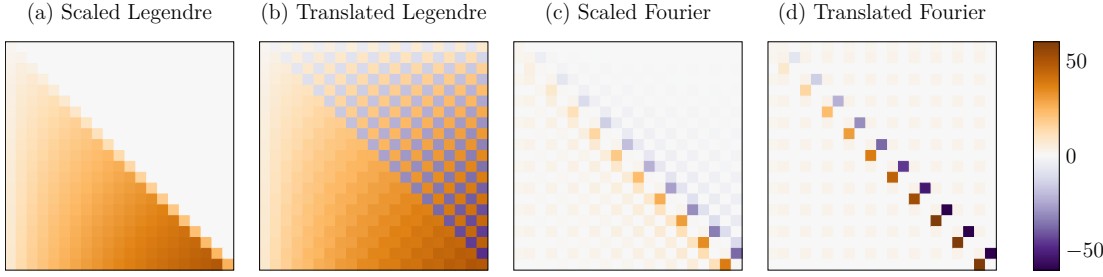

Figure 5: Examples of the $A$ matrices for several basis/measure combinations: (a) Scaled Legendre, (b) Translated Legendre, (c) Scaled Fourier, and (d) Translated Fourier. The dense non-zero elements in the upper right of (b) explain its poor performance compared to (a). The numerous small nonzero elements above the diagonal in (c) and (d) contribute to mixing errors over long sequences.

1. compatibility between the frame and the given class of input signals, and

2. the operator $A$ that results from a given frame has a small contaminating part.

Truncation and mixing errors have different sources but are linked. The optimal strategy to reduce both at the same time is to choose a basis that results in a representation where the energy is concentrated in the first $N$ coefficients. This can only be achieved, however, if we have some prior knowledge of the input signal in order to choose the right basis and truncation level. In cases where little is known about the input signal or correct truncation level, it is advisable to instead choose an SSM that is zero in the upper right quadrant, such as HiPPO-LegS, as it will inherently negate mixing errors.

### 5.2.4  Error bound

In order to quantify the mixing error, we show that the truncated representation follows the same differential equation with a perturbation defined by the theorem below.

**Theorem 3.** *(Poof in Appendix A.6) The truncated representation generated by scaled-SaFARi follows a differential equation similar to the full representation, with the addition of a perturbation factor:*

$$\frac{\partial}{\partial T}c = -\frac{1}{T}Ac + \frac{1}{T}Bu(T) - \frac{1}{T}\vec{\epsilon}(T), \tag{23}$$

*where*

$$\vec{\epsilon}(T) := \langle u_T, \xi \rangle, \qquad \xi = \Upsilon(\widetilde{\Phi}\Phi - I). \tag{24}$$

The mixing error term $\vec{\epsilon}(T)$ cannot be directly calculated; however, we can derive an upper bound for the mixing error and demonstrate that this bound can be made arbitrarily small with appropriate truncation. This generic error bound is not necessarily tight; tighter bounds could be identified for specific instantiations of SSMs for given parameters (frame or basis, measure, dimension, etc.)

**Theorem 4.** *If one finds an upper bound such that for all the times before $T$ we have $\|\epsilon(T)\|_2 < \epsilon_m$, then the representation error is bound by*

$$\|\Delta c(T)\|_2 < \epsilon_m \sqrt{\sum \frac{1}{\lambda_i^2}} = \epsilon_m \|A^{-1}\|_F \tag{25}$$

*where $\lambda_i$ are the eigenvalues of $A$, and $F$ indicates the Frobenius norm. See proof in Appendix A.6.*

### 5.2.5  Error analysis for different SaFARi$^{(N)}$ constructs

In Section 5.1.1 we define SaFARi$^{(N)}$ as a finite dimensional approximation for SaFARi, and provide two particular constructions, DoT and ToD. Armed with the quantification of representation error, we compare these different constructs. We provide Theorem 5 to demonstrate DoT has the optimal representation error between different choices for SaFARi$^{(N)}$ using the same frame.

**Theorem 5.** *Given a frame $\Phi$, the Dual of Truncation (DoT) construct introduced in Section 5.1.3 has optimal representation error when compared to any other SaFARi$^{(N)}$ construct for the same frame $\Phi$. See proof in Appendix A.6.*

As established in Theorem 5, SaFARi$^{(N)}$ SSMs should be constructed with the DoT method.

## 6  Computational and runtime complexity

This section develops the computational methods for obtaining sequence representations with SaFARi, emphasizing its efficiency in both training and inference phases. We analyze the computational complexity of different update methods, and highlight the benefit of parallel computation with diagonalizable SSMs. These discussions provide a foundation for understanding the scalability of SaFARi in practical applications.

## 6.1 Computational complexity for sequential updates

To compute representations using scaled-SaFARi, the GBT update requires solving an $N \times N$ linear system at each step, leading to $O(N^3 L)$ complexity for a sequence of length $L$. In contrast, translated-SaFARi reuses the same inverse across all steps, reducing the complexity to $O(N^2 L)$.

If the state matrix $A$ is diagonalizable, both scaled and translated variants can be accelerated. The sequence representation $\widetilde{C}$ is computed for the diagonal SSM and transformed back via $C = V\widetilde{C}$, reducing the complexity to $O(NL)$. On parallel hardware, such diagonalized systems decompose into $N$ independent scalar SSMs, yielding $O(L)$ runtime with sufficient parallel resources.

Diagonalizability of an SSM depends on the frame or basis used in its construction. One limitation of Legendre-based SSMs such as HiPPO is that its $A$ matrix cannot be stably diagonalized, even for representation sizes as small as $N = 20$ (Gu et al., 2022a), leading to significantly higher cost. To address this limitation, (Gu et al., 2023) proposed a fast sequential update method for HiPPO-LegS, claiming $O(N)$ computational complexity and $O(1)$ runtime complexity per update on parallel processors. However, we observe that this method becomes numerically unstable at larger $N$, as discussed in Appendix A.7.1. To resolve this, we suggest a simple modification: computing lower-degree representations before higher-degree ones. While the modified approach preserves the $O(N)$ overall computational cost, it increases runtime to $O(N)$ per sequential update, as it is no longer parallelizable.

Similar to HiPPO-LegS for the scaled measure, HiPPO-LegT for the translated measure also cannot be stably diagonalized. To our knowledge, LegT has no analogue for the fast LegS update method in Gu et al. (2020). Therefore, the computational complexity remains considerably higher than for diagonalizable SSMs.

## 6.2 Convolution kernels and diagonalization

When using an SSM for a recognition or learning task, a training phase is required in which the downstream model is trained on the generated representation. Using sequential updates for training is prohibitively taxing on computational and time resources as the whole sequence is available in the training time. Ideally, we would perform computations of the sequential SSM in parallel. However, this is a challenge since each new update depends on the result of the previous. The authors of Gu et al. (2022b) discussed how to implement a parallel computation algorithm for SSMs produced by HiPPO. To do so, one "unrolls" the SSM as:

$$c_k = \overline{A}_k \ldots \overline{A}_1 \overline{B}_0 u_0 + \cdots + \overline{A}_k \overline{B}_{k-1} u_{k-1} + \overline{B}_k u_k \quad, \quad c = K * u \tag{26}$$

$$\overline{A}_i = (I + \delta t \alpha A_i)^{-1}(I - \delta t(1-\alpha)A_i), \qquad \overline{B}_i = (I + \delta t \alpha A_{i+1})^{-1} B_i. \tag{27}$$

The convolution kernel $K$ in Eq. 26 removes the sequential dependency, enabling parallel computation on hardware such as GPUs, and significantly reducing training time.

## 6.3 Runtime complexity

### 6.3.1 Runtime complexity of scaled-SaFARi

If the SSM is not diagonalizable, then the kernel can still be computed in parallel by framing the problem as a scan over the associative prefix product. Since matrix multiplication is associative, all such prefix products can be computed efficiently using parallel scan algorithms (Blelloch, 1990; Hillis & Jr., 1986). When implemented on parallel hardware such as GPUs, this strategy achieves a time complexity of $O(N^3 \log L)$ if enough parallel processors are available.

However, if the SSM is diagonalizable, all the matrix products $\bar{A}_k$ matrices in the matrix products appearing in the kernel expression become diagonal. As a result, the convolution kernel can be calculated with the time complexity of $O(N \log L)$. Furthermore, the below theorem suggests how to find the kernel in closed form.

**Theorem 6.** *For scaled-SaFARi, if $A$ is diagonalizable, then the convolution kernel $K$ that computes the representations can be found in closed form. See Appendix A.7 for the closed-form solution.*

| Name | Spans L2 | Orthogonal | Redundant | Category |
|------|----------|------------|-----------|----------|
| Legendre | ✓ | ✓ | ✗ | Orthonormal Basis |
| Laguerre | ✓ | ✓ | ✗ | Orthonormal Basis |
| Chebyshev | ✓ | ✓ | ✗ | Orthonormal Basis |
| Fourier | ✓ | ✓ | ✗ | Orthonormal Basis |
| Bernstein | ✓ | ✗ | ✗ | Non-orthogonal Basis |
| Gabor | ✓ | ✗ | ✓ | Frame |
| Random Harmonics | ✗ | ✓ | ✗ | Neither |
| Fourier+Random Harmonics | ✓ | ✗ | ✓ | Frame |

Table 2: The first four elements in this table were studied in previous works (Gu et al. (2020; 2023)). We reproduce them here for comparison. We also introduce several new variants with characteristics such as non-orthogonality and redundancy, which can now be handled with SaFARi.

As noted above, HiPPO-LegS is not diagonalizable, complicating kernel computation. A heuristic method proposed by Gu et al. (2023) enables approximate kernel evaluation with time complexity $O(N \log L)$, but it introduces additional computation and lacks a closed-form solution.

### 6.3.2 Runtime complexity of translated-SaFARi

Similar to the scaled case, all the matrix products $\bar{A}_k \ldots \bar{A}_{k-m}$ can be computed efficiently using parallel scan algorithms (Blelloch, 1990; Hillis & Jr., 1986). As a result, the convolution kernel $K$ can be computed with an overall time complexity of $O(N^2 \log(L))$ since $\bar{A}_k$ remains the same for all values of $k$. Similarly to the scaled case, if the $A$ matrix is diagonalizable, then the $\bar{A}$ matrices become diagonal. With access to parallel processors, the runtime complexity can be reduced to $O(N^2 \log(L))$.

Furthermore, for diagonalizable SSMs, the convolution kernel for the translated measure has a closed form.

**Theorem 7.** *For translated-SaFARi, if A is diagonalizable, then the convolution kernel K that computes the representations can be found in closed form. See Appendix A.7.*

## 7 Empirical validation

We demonstrate that SaFARi can generate SSMs for function approximation over any frame or basis by choosing examples that are non-orthogonal, incomplete, or redundant. We then evaluate SaFARi-generated state-space models on some sample datasets online function approximation, benchmarking against established baselines. Code to replicate the results of this section, as well as generate SSMs with arbitrary frames is provided at: `https://github.com/echbaba/safari-ssm`.

### 7.1 Instantiations with different frames

No single frame is universally optimal all input classes. Different signal families exhibit different decay rates in representation error depending on the chosen frame or basis. We instantiate SaFARi over several sets of functions as in Fig. 6, which may constitute bases, frames, or neither, as summarized in Table 2. Description and further discussion can be found in Appendix A.8. For each function family, we scale the functions to an interval of $[0, 1]$.

### 7.2 Diagonalization of $A$

As discussed in Sec. 6.3, not all bases produce an $A$ matrix that is stably diagonalizable. This section does not attempt to provide rules to guarantee a stably diagonalizable $A$. However, we consider two key metrics that illustrate how the choice of basis and measure can result in an $A$ matrix with more (or less) desirable properties: how sensitive eigendecomposition is to perturbation, and the effective rank of $A$.

From the Bauer-Fike theorem (Bauer & Fike (1960)), given a matrix $A$ with eigenvalues $\lambda_i$ and eigenvectors $V$, and a perturbed matrix $\widetilde{A} = A + E$ with corresponding eigenvalues $\widetilde{\lambda}$, we have that[2]

$$|\widetilde{\lambda} - \lambda| \leq \kappa(V) \left\| E \right\|_2, \tag{28}$$

where $\kappa(V) = \|V\|_2\|V^{-1}\|_2$. This theorem describes a relationship between a matrix perturbation and the impact on its eigenvalues, which has a bound determined by the condition number of the eigenvectors of $A$. The larger this value, the less numerically stable the operations on $A$ will be. Varying $N$ for different SaFARi instantiations can result in unstable diagonalization of $A$ matrices, as shown in Fig. 7.

$\kappa(V)$ gives some information about the stability of the eigenvalue decomposition, but does not address structural issues such as degenerate eigenvalues. To address this, we consider the effective rank, defined by Roy & Vetterli (2007) as the Shannon entropy of the normalized singular values $p_k$. The closer this value is to $N$, the less redundancy of eigenvalues.

$$\text{erank} = \exp\left(-\sum_{k=1}^{N} p_k \log p_k\right) \tag{29}$$

While these metrics may not account for all possible edge cases, in general, we would like the $A$ matrix to have a very low value for $\kappa(V)$ and a high value for erank. Notably, in Fig. 7, several of the orthogonal polynomials of prior work are suboptimal, whereas some of our newly-introduced frames (which can only be constructed via SaFARi, having no closed-form solution) show improvements in both metrics.

### 7.3 Datasets for experiments

**S&P 500**: We use the daily S&P 500 index as a broad, large-cap U.S. equities benchmark over the last decade: from August 2015 to August 2025 (Yahoo Finance (2025)). The series consists of end-of-day levels for the price index. Overlapping sequences of 500 samples are collected as different instances of time series to form a dataset, and resampled into 4,000 samples to emulate longer time-series signals.

---

[2]The norm $\| \cdot \|$ in Eq. 28 can be any p-norm, e.g. 1, 2, or $\infty$. The 2-norm was used here, which denotes the largest singular value of E.

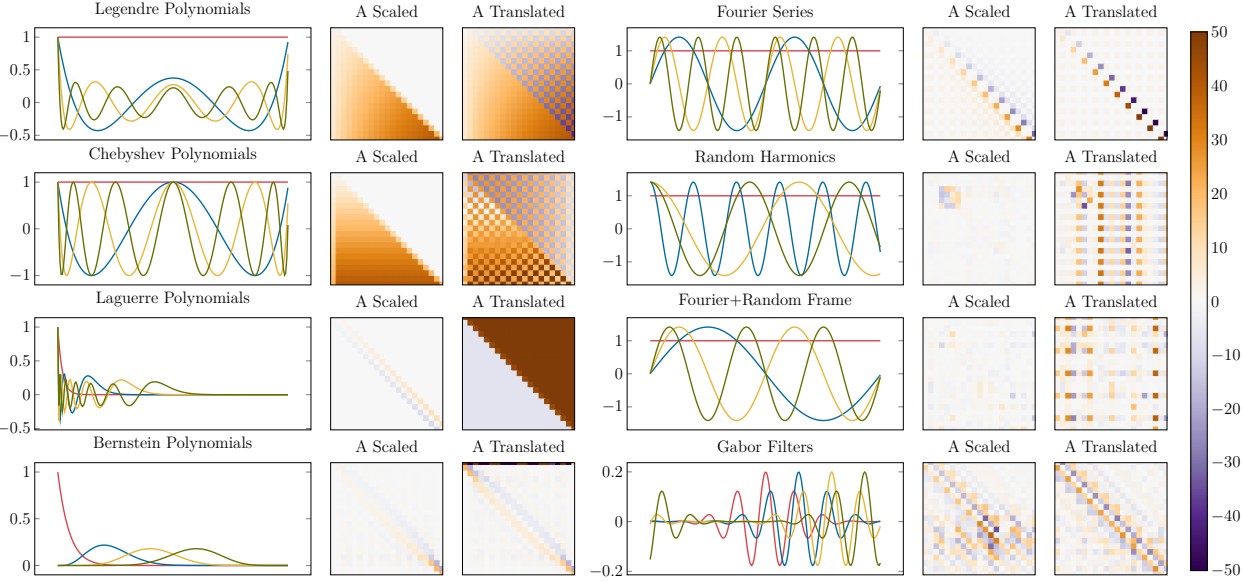

Figure 6: For each of the SSMs instantiated in this work, we show a few elements of the frame, and the resulting $A$ matrices for scaled and translated measures. The scale for the $A$ matrices is the same as in Fig. 5 for consistency.

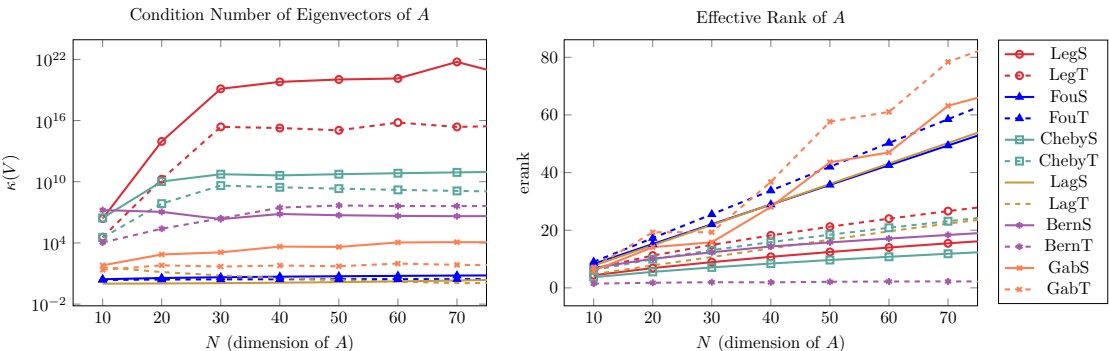

Figure 7: Comparison of $A$ matrices produced by different frames via SaFARi. The condition number of eigenvectors of $A$ indicates how sensitive the diagonalization is to perturbations, and the effective rank relates to the distribution of eigenvalues. Our results support the findings of prior work in Gupta et al. (2024); Gu et al. (2022a), noting that at relatively low $N$, Legendre-based $A$ matrices rapidly become difficult to stably diagonalize, but other frames (such as Fourier and Gabor) do not.

**M4**: The M4 dataset consists of a collection of time-series data across different realms, including economic, financial, industrial, and demographic, at various intervals.

### 7.3.1 Function memorization and approximation

Members of the M4 and S&P 500 datasets were passed to SSMs constructed from different bases and frames. At each iteration, the SSM generates a coefficient vector $c$ (see Fig. 1, Eq. 13 and subsequent equations) which is a compressed representation of the time-series signal up to that point. The error between the reconstructed signal from $c$ and the original input signal is a measure of how effectively the SSM can remember and represent the signal. An illustration based on a single instance of M4 is shown in Fig. 8.

**Scaled and translated SSMs:** For the translated case, the window size is set at 10% of the input signal length. Since we are only estimating a small portion of the signal each time we evaluate the translated case, the MSE reported is the mean of all intermediate estimates. For consistency, the error in the scaled case is measured the same way. The MSEs between the reconstructed signal and the estimated signal are reported in Table 3.

**Size and Rank:** Both scaled and translated versions were evaluated with $N = 32, 64, 128$, where $N$ is the size of the signal representation. For orthonomal bases, basis elements are lin-

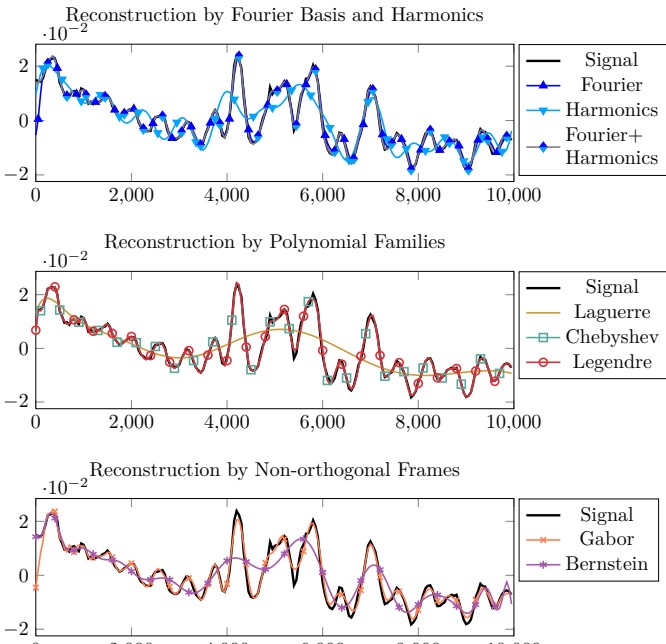

Figure 8: One sample from the M4 dataset reconstructed by different SaFARi implementations. These are separated into multiple subfigures for visibility.

early independent, so both basis and $A$ matrix have rank $N$. For frames, elements have redundancy, so the SSM has effective rank $N_{\text{eff}} < N$. In Table 3, the two frames (Fourier+Random Harmonics and Gabor) are treated separately to highlight the distinction. We first instantiate these frames with $N$ member vectors (Fou+, Gabor). We then add $N^* > N$ member vectors and diagonalize the resulting $A$ matrix, so that the resulting ($N_{\text{eff}}$) matches the desired $N$ (Fou+*, Gabor*).

| | Scaled | | | | | | Translated | | | | | |
| | N = 32 | | N = 64 | | N = 128 | | N = 32 | | N = 64 | | N = 128 | |
| | SP | M4 | SP | M4 | SP | M4 | SP | M4 | SP | M4 | SP | M4 |
|---|---|---|---|---|---|---|---|---|---|---|---|---|
| Leg | 0.039 | 0.328 | 0.039 | 0.322 | 0.039 | 0.322 | 0.166 | 0.288 | 0.166 | 0.287 | 0.166 | 0.289 |
| Fou | 0.018 | 0.218 | 0.013 | 0.159 | 0.009 | 0.109 | 0.314 | 0.271 | 0.314 | 0.270 | 0.314 | 0.271 |
| Lag | 0.678 | 0.712 | 0.678 | 0.709 | 0.678 | 0.712 | 0.313 | 0.271 | 0.314 | 0.207 | – | – |
| Cheby | 0.015 | 0.162 | 0.013 | 0.117 | 0.012 | 0.077 | 0.021 | 0.179 | 0.024 | 0.176 | 0.030 | 0.181 |
| Bern | 0.026 | 0.194 | 0.020 | 0.173 | 0.017 | 0.158 | 0.021 | 0.179 | 0.021 | 0.174 | 0.021 | 0.178 |
| Rand | 0.026 | 1.175 | 0.020 | 0.563 | 0.014 | 0.249 | 0.022 | 0.179 | 0.023 | 0.176 | 0.026 | 0.179 |
| Fou+ | 0.017 | 0.191 | 0.011 | 0.152 | 0.009 | 0.109 | 0.021 | 0.179 | 0.023 | 0.175 | 0.026 | 0.180 |
| Fou+* | 0.013 | 0.156 | 0.009 | 0.101 | 0.007 | 0.058 | 0.022 | 0.179 | 0.025 | 0.176 | 0.027 | 0.181 |
| Gabor | 0.024 | 0.266 | 0.018 | 0.191 | 0.714 | 0.206 | 0.021 | 0.179 | 0.022 | 0.175 | 0.023 | 0.179 |
| Gabor* | 0.018 | 0.223 | 0.016 | 0.181 | 0.051 | 0.144 | 0.022 | 0.179 | 0.022 | 0.175 | 0.024 | 0.179 |

Table 3: MSE of reconstructed signals with different instantiations of SaFARi. The table is divided into polynomial (top) and non-polynomial (bottom) representations. Asterisks indicate that the frame was augmented with additional vector elements to achieve an effective rank. For each test, the minimum MSE for polynomial and non-polynomial SSMs is shaded. Missing entries for Laguerre could not be computed due to numerical errors arising from exponents in higher-order terms. The standard deviation is reported in Appendix A.9.

**Discussion:** In Table 3, the lowest MSE depends on a combination of $N$, the windowing function, and the data being processed. However, some broad patterns emerge. Most of the best-performing SSMs have characteristics of redundancy (Fou+, Fou+*, Gabor, Gabor*) or non-orthogonality (Bernstein), which were never explored in prior work due to their lack of a closed-form solution for $A$ and $B$. This suggests that alternative SSMs constructed via SaFARi could improve the performance of SSM-based models.

We also observe that the representative power of a given SSM depends in part on the length of the signal under consideration. In the translated case, the window size is constant, so any SSM that can adequately model features in a signal of length $W$ will have negligible performance advantages. In the scaled case, however, the length of the signal changes at each iteration, and the choice of frame or basis for SSM can have a significant impact on MSE.

Ultimately, the choice of basis or frame is task-dependent, and different representation structures will perform better or worse, depending on the signal's features. No single SSM is universally optimal for all signal types. However, given our observations regarding redundancy and non-orthogonality, wavelets are a natural choice. It was found in Babaei et al. (2025) that Daubechies wavelets can generally outperform polynomial-based HiPPO SSMs for time-series signals, especially when those signals contain discontinuities or other non-idealities.

### 7.3.2 Comparing SSMs to learned structures in the long horizon delay task

To compare the performance of SaFARi SSMs as online memory units, SaFARi features are computed on a signal by sequential update, then the observed scalar input at $t = T - d$ (where $T$ is the current time sample and $d$ is the delay) earlier is reconstructed from the representation via the dual frame. The MSE between the reconstruction and the ground truth delayed input is an indicator of the memory performance of the structure, as it demonstrates that information in the long history horizons is recoverable. For each test, we use the same delay amount and the same representation size of 32. The SSM-based models in our study do not include any learned parameters; their dynamics are fully determined by the state-space formulation, so their number of *learnable* parameters is effectively zero. The LSTM and GRU models are trained using an Adam optimizer (Kingma (2014)) until they converge, and the final validation performance is shown in Fig. 9.

**Discussion:** The trends in Fig. 9 reflect how SSM-based models demonstrate better memory performance than learnable models for the same representation size. Models with learnable parameters (LSTMs, GRUs)

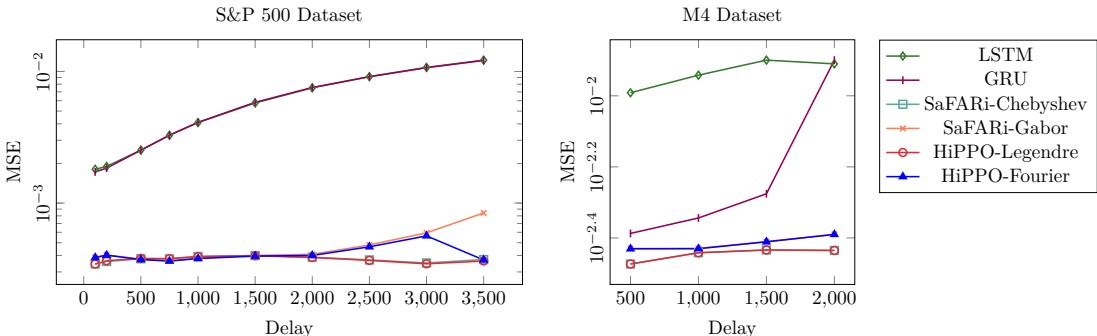

Figure 9: Memory recall performance of learned models, HiPPO SSMs, and SaFARi SSMs in the long horizon delay task. Legendre and Chebyshev SSMs are nearly indistinguishable. Gabor and Fourier also track closely.

need large amounts of data to generalize well. When the delay horizon is long and the available data is limited, these models tend to perform worse. This is consistent with our observation that the gap between RNNs and SSMs is smaller for the M4 dataset, which has many more samples, compared to the S&P 500 dataset. Even in data-rich settings, RNNs prioritize short-term dependencies because of their internal gating mechanisms and training dynamics, which helps avoid vanishing or exploding gradients. This bias limits their ability to retain information from long time horizons. In contrast, SSM models are designed to maintain information flow over longer sequences.

Another notable trend in the plots of Fig 9 is the varying MSE as the delay increases. Intuitively, at longer delays, the memorization task becomes more difficult, and we expect the MSE to increase. The MSE does increase overall, but not strictly monotonically, and this is especially salient for the SSM models. If the underlying time series exhibits non-monotonic autocorrelation structure, then periodic or quasi-periodic components cause the signal at specific lags to be more correlated (and thus more predictable) than at nearby lags, which is often the case for real-world signals Kantz & Schreiber (2003). SSM models also contain structured functions that exhibit periodicity (see Fig. 6), whereas learned models do not necessarily converge to periodic representations. Thus, the same structure that gives SSMs an advantage in MSE performance can also make them more susceptible to periodic correlation in delay and copying tasks.

## 8 Conclusions

In this work, we demonstrate how our method, SaFARi, generalizes HiPPO (Gu et al. (2020)) to accommodate bases or frames where closed-form solutions are unavailable, paving the way for broader applicability in sequential modeling tasks. Key findings of this work regarding the choice of frame or basis of an SSM motivate the need for more flexibility than prior methods could provide:

- The underlying frame of the SSM should be **compatible with the signal of interest**; there is no "one size fits all". The correct choice of frame reduces both the error and representation size (Sec. 7).

- Notably, SSMs using **non-orthogonal and redundant frames**—which do not have closed-form solutions—can outperform the standard polynomial-based instances (Sec. 7).

- Even for an optimal frame, an SSM's performance will also depend on **structural features of the $A$ matrix** (Sec. 5.2). Different frames result in $A$ matrices with better or worse **numerical properties** (Sec. 7.2), and **stable diagonalization of the $A$ matrix** is critical for computational efficiency (Sec. 6.3).

The groundwork laid in this paper lends itself naturally to several future research directions. One considers SaFARi as a standalone representation module. While we have presented several new frames and bases in this work, there are many other structured frames, such as different wavelet types, that could be leveraged for localized and sparse representations.

Another direction focuses on the integration of SaFARi into learned models, specifically advanced SSM architectures such as S4 and Mamba. By embedding known temporal structures of dynamical systems directly into the SSM architecture with SaFARi, we can improve memory and reconstruction performance of the core SSM, which in turn reduces the need to learn all parameters from scratch. We anticipate that this will reduce the computational burden of training, while increasing stability and parameter efficiency. SaFARi is also a natural candidate for implementing gating-like behavior: the memory encoded in the state vector determines how historical information is selectively integrated into downstream computations. In this sense, SaFARi modules could function analogously to gating layers in RNNs, but with an efficiently designed, context-aware structure.

SaFARi also enables us to investigate the synergy between ML models and SSM variants. The interplay between the internal structure of the SSM (from the frame and measure) and the architecture of the learned model that contains it may play an important role in overall performance. For example, a wavelet-based SSM that detects discontinuities in signals may have better synergy with models designed for sparse data, whereas polynomial-based SSMs techniques may be better suited for use with techniques involving logistic regression.

By extending SSM construction beyond specific bases, SaFARi provides a flexible foundation for efficient state-space representations by linking theoretical advancements to practical applications in sequential data processing.

## Acknowledgments

The authors thank T. Mitchell Roddenberry for fruitful conversations over the course of this project, and Matt Ziemann for assistance with code development. This work was supported by NSF grants CCF-1911094, IIS-1838177, and IIS-1730574; ONR grants N00014-18-12571, N00014-20-1-2534, N00014-18-1-2047; MURI N00014-20-1-2787; AFOSR grant FA9550-22-1-0060; and DOE grant DE-SC0020345. Additional support came from a Vannevar Bush Faculty Fellowship, the Rice Academy of Fellows, and the Rice University and Houston Methodist 2024 Seed Grant Program.

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

# A  Appendix

## A.1  Derivation of SaFARi with the scaled measure

**Theorem** (1)**.** *Assuming elements in the frame $\Phi$ and the input signal $u$ are right-continuous, for the representation defined in Eq. 14, the partial derivative is:*

$$\frac{\partial}{\partial T}\vec{c}(T) = -\frac{1}{T}A\vec{c}(T) + \frac{1}{T}Bu(T) \tag{A.1}$$

*where $A$ is a linear operator defined as:*

$$A = I + \mathbb{U}_\Upsilon \mathbb{U}_{\widetilde{\Phi}}^* \tag{A.2}$$

*and $B$ is the complex conjugate of a vector containing members of the main frame evaluated at $T = 1$ so we have $B = \{\overline{\phi}_n(T=1)\}_{n\in\Gamma}$ . One can show that the $A$ operator can also be described using the matrix multiplication:*

$$A_{i,j} = \delta_{i,j} + \int_0^1 \overline{v}_i(t)\widetilde{\phi}_j(t)dt \tag{A.3}$$

**Proof**: Taking partial derivative with respect to $T$, we have:

$$\frac{\partial}{\partial T}c_n(T) = \frac{\partial}{\partial T}\int_{t=0}^{T}\frac{1}{T}u(t)\overline{\phi}_n\left(\frac{t}{T}\right)dt \tag{A.4}$$

Now applying the Leibniz integral rule we have:

$$\begin{aligned}
\frac{\partial}{\partial T}c_n(T) &= \int_{t=0}^{T}\frac{-1}{T^2}u(t)\overline{\phi}_n\left(\frac{t}{T}\right)dt + \int_{t=0}^{T}\frac{-1}{T^2}u(t)\frac{t}{T}\overline{\phi'_n}(\frac{t}{T})dt + \frac{\overline{\phi}_n(1)}{T}u(T) \\
&= \frac{-1}{T}c_n + \frac{-1}{T}\int_{t=0}^{T}u(t)\frac{1}{T}\overline{v}_n\left(\frac{t}{T}\right)dt + \frac{\overline{\phi}_n(1)}{T}u(T) \\
&= \frac{-1}{T}c_n + \frac{-1}{T}\langle u, v_n\rangle + \frac{\overline{\phi}_n(1)}{T}u(T)
\end{aligned} \tag{A.5}$$

This is still not an SSM since the second term is not explicitly a linear form of $\vec{c}(T)$. To convert this to a linear form of $\vec{c}(T)$, we use the equality given in Eq. 11 to represent $v_n$ using the frame $\widetilde{\Phi}$

$$v_n(t) = \sum_{j\in\Gamma}\langle v_n, \phi_j\rangle\widetilde{\phi}_j = \sum_{j\in\Gamma}\langle v_n, \widetilde{\phi}_j\rangle\phi_j \tag{A.6}$$

$$\rightarrow \langle u, v_n \rangle = \langle u, \sum_{j \in \Gamma} \langle v_n, \widetilde{\phi}_j \rangle \phi_j \rangle = \sum_{j \in \Gamma} \overline{\langle v_n, \widetilde{\phi}_j \rangle} \langle u, \phi_j \rangle = \sum_{j \in \Gamma} \overline{\langle v_n, \widetilde{\phi}_j \rangle}\, c_j(T) \tag{A.7}$$

Putting Eq. A.7 in Eq. A.5 results in:

$$\frac{\partial}{\partial T} \vec{c}(T) = -\frac{1}{T}(I + \mathbb{U}_\Upsilon \mathbb{U}_{\widetilde{\Phi}}^*) \vec{c}(T) + \frac{\overline{\phi}_n(1)}{T} u(T) \tag{A.8}$$

This proves the theorem. $\square$

## A.2 Derivation of SaFARi with the translated measure

**Theorem** (2). *Assuming elements in the frame $\Phi$ and the input signal $u$ are right-continuous, for the representation defined in Eq. 14, the partial derivative is:*

$$\frac{\partial}{\partial T} \vec{c}(T) = -\frac{1}{\theta} A \vec{c}(T) + \frac{1}{\theta} B u(T) \tag{A.9}$$

*where $A$ is a linear operator defined as:*

$$\overline{A} = \mathbb{U}_{\dot{\Phi}} \mathbb{U}_{\widetilde{\Phi}}^* + \mathbb{Q}_\Phi \mathbb{Q}_{\widetilde{\Phi}}^* \tag{A.10}$$

*and $B$ is the complex conjugate of a vector containing members of the main frame evaluated at $T = 1$ so we have $B = \{\overline{\phi}_n(T = 1)\}_{n \in \Gamma}$. One can show that the $A$ operator can also be described using the matrix multiplication:*

$$A_{i,j} = \overline{\phi}_i(0) \widetilde{\phi}_j(0) + \int_0^1 \left[ \frac{\partial}{\partial t} \overline{\phi}_i(t) \right]_{t=t'} \widetilde{\phi}_j(t') dt' \tag{A.11}$$

**Proof**: We can write the coefficients as:

$$c_n(T) = \int_{T-\theta}^T u(t) \frac{1}{\theta} \overline{\phi}_n \left( \frac{t - (T - \theta)}{\theta} \right) dt \tag{A.12}$$

Taking the partial derivative with respect to $T$, we have

$$\frac{\partial c_n(T)}{\partial T} = \frac{-1}{\theta^2} \int_{T-\theta}^T u(t) \overline{\phi}_n' \left( \frac{t - (T - \theta)}{\theta} \right) dt + \frac{1}{\theta} \overline{\phi}_n(1) u(T) - \frac{1}{\theta} \overline{\phi}_n(0) u(T - \theta) \tag{A.13}$$

Similar to our approach for the previous theorem, we write $\phi_n'(z)$ as

$$\phi_n'(z) = \sum_{i \in \Gamma} \langle \phi_n', \widetilde{\phi}_i \rangle \phi_i(z) = \sum_{i \in \Gamma} Q_{n,i}\, \phi_i(z) \tag{A.14}$$

$$Q_{n,i} = \int_{z=0}^1 \phi_n'(z) \overline{\widetilde{\phi}}_i(z) dz \tag{A.15}$$

Now, if we use this expansion, and put it in Eq. A.13 we have

$$\frac{\partial c_n(T)}{\partial T} = \frac{-1}{\theta^2} \int_{T-\theta}^T u(t) \left[ \sum_{i \in \Gamma} \overline{Q}_{n,i}\, \overline{\phi}_i \left( \frac{t - (T - \theta)}{\theta} \right) \right] dt + \frac{1}{\theta} \overline{\phi}_n(1) u(T) - \frac{1}{\theta} \overline{\phi}_n(0) u(T - \theta) \tag{A.16}$$

$$\frac{\partial c_n(T)}{\partial T} = \frac{-1}{\theta} \sum_{i \in \Gamma} \overline{Q}_{n,i} \left[ \frac{-1}{\theta} \int_{T-\theta}^T u(t) \overline{\phi}_i \left( \frac{t - (T - \theta)}{\theta} \right) dt \right] + \frac{1}{\theta} \overline{\phi}_n(1) u(T) - \frac{1}{\theta} \overline{\phi}_n(0) u(T - \theta) \tag{A.17}$$

We also write $u(T - \theta)$ as a reconstruction using the current representation $u(T - \theta) = \sum_i c_i \widetilde{\phi}_i(0)$

$$\frac{\partial c_n(T)}{\partial T} = \frac{-1}{\theta} \sum_i \overline{Q}_{n,i}\, c_i + \frac{1}{\theta} \overline{\phi}_n(1) u(T) - \frac{1}{\theta} \overline{\phi}_n(0) \sum_i c_i \widetilde{\phi}_i(0) \tag{A.18}$$

$$\frac{\partial c_n(T)}{\partial T} = \frac{-1}{\theta} \sum_i (\overline{Q}_{n,i} + \widetilde{\phi}_i(0) \overline{\phi}_n(0))\, c_i + \frac{1}{\theta} \overline{\phi}_n(1) u(T) \tag{A.19}$$

If we put $A_{i,j} = \overline{Q}_{i,j} + \overline{\phi}_i(0) \widetilde{\phi}_j(0)$, it proves the theorem. $\square$

### A.3 Mathematical properties of SaFARi

**Proposition.** *For any scalar $\beta > 0$, if $h(t) = u(\beta t)$ then for the scaled measure we have* SaFARi(h)(T) = SaFARi(u)($\beta$T)

**Proof**: We start by writing the representation generated by the scaled SaFARi for $h(t)$.

$$\text{SaFARi(h)(T)} = \int_{t=0}^{T} h(t)\overline{\phi_n}\left(\frac{t}{T}\right)\frac{1}{T}dt \tag{A.20}$$

$$= \int_{t=0}^{T} u(\beta t)\overline{\phi_n}\left(\frac{t}{T}\right)\frac{1}{T}dt \tag{A.21}$$

$$\int_{t=0}^{T} u(t')\overline{\phi_n}\left(\frac{t'}{\beta T}\right)\frac{1}{T}\frac{dt'}{\beta} = \text{SaFARi(u)}(\beta T) \tag{A.22}$$

**Proposition.** *For any scalar $\beta > 0$, if $h(t) = u(\beta t)$ then for the translated measure with parameter $\theta$ we have* SaFARi$_\theta$(h)(T) = SaFARi$_{\beta\theta}$(u)($\beta$T)

Similar to the previous proposition, we start by writing the representation of the scaled SaFARi:

$$\text{SaFARi}_\theta(\text{h})(\text{T}) = \int_{t=T-\theta}^{T} h(t)\overline{\phi_n}\left(\frac{t-(T-\theta)}{\theta}\right)\frac{1}{\theta}dt \tag{A.23}$$

$$= \int_{t=T-\theta}^{T} u(\beta t)\overline{\phi_n}\left(\frac{t-(T-\theta)}{\theta}\right)\frac{1}{\theta}dt \tag{A.24}$$

$$= \int_{t'=\beta T-\beta\theta}^{\beta T} u(t')\overline{\phi_n}\left(\frac{t'-(\beta T-\beta\theta)}{\beta\theta}\right)\frac{1}{\theta}\frac{dt'}{\beta} = \text{SaFARi}_{\beta\theta}(\text{u})(\beta\text{T}) \tag{A.25}$$

### A.4 The closed-form solution for SaFARi differential equations

**Lemma 1.** *The closed form solution for the differential equation introduced in Eq. 15 is:*

$$c = \int_{t=0}^{T} \frac{1}{t}\exp\left(A\ln\frac{t}{T}\right)Bu(t)dt. \tag{A.26}$$

**Proof**: We begin by re-writing the differential equation for any time $t$

$$\frac{\partial}{\partial t}\vec{c}(t) + \frac{1}{t}A\vec{c}(t) = \frac{1}{t}Bu(t) \tag{A.27}$$

Now we multiply both sides by $\exp(A\ln(t))$

$$\exp(A\ln(t))\frac{\partial}{\partial t}\vec{c}(t) + \frac{1}{t}A\exp\left(A\ln(t)\right)\vec{c}(t) = \frac{1}{t}\exp(A\ln(t))Bu(t) \tag{A.28}$$

The left side of the equality is now a complete differential

$$\frac{\partial}{\partial t}\left(\exp(A\ln(t))\vec{c}(t)\right) = \frac{1}{t}\exp\left(A\ln(t)\right)Bu(t) \tag{A.29}$$

$$\exp\left(A\ln(T)\right)\vec{c}(T) = \int_{t=0}^{T} \frac{1}{t}\exp\left(A\ln(t)\right)Bu(t) \tag{A.30}$$

$$\vec{c}(T) = \exp\left(-A\ln(T)\right)\int_{t=0}^{T} \frac{1}{t}\exp\left(A\ln(t)\right)Bu(t) \tag{A.31}$$

This proves Lemma 1. $\square$

**Lemma 2.** *The closed form solution for the differential equation introduced in Eq. 19 is:*

$$c = \int_{t=T-\theta}^{T} \frac{1}{\theta} \exp\left(A\frac{t-T}{\theta}\right) Bu(t). \tag{A.32}$$

**Proof**: We begin by re-writing the differential equation for any time $t$ as

$$\frac{\partial}{\partial t}\vec{c}(t) + \frac{1}{\theta}A\vec{c}(t) = \frac{1}{\theta}Bu(t). \tag{A.33}$$

Now we multiply both sides by $\exp(A\frac{t}{\theta})$

$$\exp\left(A\frac{t}{\theta}\right)\frac{\partial}{\partial t}\vec{c}(t) + \frac{1}{\theta}A\exp\left(A\frac{t}{\theta}\right)\vec{c}(t) = \frac{1}{\theta}\exp\left(A\frac{t}{\theta}\right)Bu(t) \tag{A.34}$$

The left side of the equality is now a complete differential

$$\frac{\partial}{\partial t}\left(\exp\left(A\frac{t}{\theta}\right)\vec{c}(t)\right) = \frac{1}{\theta}\exp\left(A\frac{t}{\theta}\right)Bu(t) \tag{A.35}$$

$$\vec{c}(T) = \exp\left(-A\frac{T}{\theta}\right)\int_{t=T-\theta}^{T} \frac{1}{\theta}\exp\left(A\frac{t}{\theta}\right)Bu(t) \tag{A.36}$$

This proves Lemma 2. $\square$

### A.5 Truncation of frames

For a sequence of matrix operators to converge, there are different types of convergence, including operator norm convergence, Strong Operator Theory (SOT) convergence, and entry-wise convergence. A sequence of operators $A_n$ is said to converge to $A$ in the Strong Operator convergence sense if $A_n \to A$ strongly if $\forall x \in l^2$:

$$\|A_n P_n x - Ax\| \to 0 \tag{A.37}$$

Where $P_n$ projects the vector $x$ into a vector containing only its first $n$ coordinates.

This type of convergence guarantees that for the SSM updates, $\|A_n\vec{c}(T) - A\vec{c}(T)\| \to 0$. This means that the difference between the true infinite dimensional update and the update using truncated $A$ vanishes to have zero norm-2, guaranteeing that the representation error using SaFARi updates can be diminished to an arbitrarily small value.

Our implementations and empirical evidence support that the DoT and ToD constructs in Section 5.1 are SaFARi$^{(N)}$ sequences. The fact that our DoT constructs reproduce the closed-form HiPPO derivations for $A$ and $B$ also provide strong evidence for this framing of the construction. However, a rigorous proof that that the two introduced structures meet the SOT convergence criteria is still needed, and should be addressed in follow-up work.

### A.6 Error analysis

**Theorem** (3). *The truncated representation generated by the scaled-SaFARi follows a differential equation similar to the full representation, with the addition of a perturbation factor.*

$$\frac{\partial}{\partial T}c = -\frac{1}{T}Ac + \frac{1}{T}Bu(T) - \frac{1}{T}\vec{\epsilon}(T). \tag{A.38}$$

*where $\vec{\epsilon}$ is defined as*

$$\vec{\epsilon}(T) = \langle u_T, \xi \rangle \tag{A.39}$$

$$\xi = \Upsilon(\widetilde{\Phi}\Phi - I) \tag{A.40}$$

**Proof**: Repeat the steps taken in the proof of Theorem 1 until Eq. A.6. Truncating the frame results in an error in this step which can be written as

$$v_n(t) = \sum_{j \in \Gamma} \langle v_n, \widetilde{\phi}_j \rangle \phi_j + \xi_n(t) \tag{A.41}$$

In fact, this is how $\xi$ is defined. Adding $\xi$ as a correction term here changes the SSM derivation:

$$\to \langle u, v_n \rangle = \langle u, \xi_n + \sum_{j \in \Gamma} \langle v_n, \widetilde{\phi}_j \rangle \phi_j \rangle = \sum_{j \in \Gamma} \overline{\langle v_n, \widetilde{\phi}_j \rangle} \langle u, \phi_j \rangle + \langle u, \xi_n \rangle = \sum_{j \in \Gamma} \overline{\langle v_n, \widetilde{\phi}_j \rangle} c_j(T) + \langle u, \xi_n \rangle \tag{A.42}$$

Putting Eq. A.42 in Eq. A.5 results in:

$$\frac{\partial}{\partial T} c = -\frac{1}{T} Ac + \frac{1}{T} Bu(T) - \frac{1}{T} \vec{\epsilon}(T). \qquad \square \tag{A.43}$$

**Theorem** (4). *If one finds an upper bound such that $\forall t < T$ we have $\|\epsilon(t)\|_2 < \epsilon_m$, then the representation error can be bound by*

$$\|\Delta c(T)\|_2 < \epsilon_m \sqrt{\sum \frac{1}{\lambda_i^2}} = \epsilon_m \|A^{-1}\|_F \tag{A.44}$$

**Proof**: Using the result of Theorem 3:

$$\frac{\partial}{\partial T} c = -\frac{1}{T} Ac + \frac{1}{T} (Bu(T) - \vec{\epsilon}(T)) \tag{A.45}$$

We can use Lemma 1 to find the closed form solution of the perturbed SSM above

$$c = \int_{t=0}^{T} \frac{1}{t} \exp\left(A \ln \frac{t}{T}\right) (Bu(t) - \vec{\epsilon}(t)) \, dt \tag{A.46}$$

$$c = \int_{t=0}^{T} \frac{1}{t} \exp\left(A \ln \frac{t}{T}\right) Bu(t) dt - \int_{t=0}^{T} \frac{1}{t} \exp\left(A \ln \frac{t}{T}\right) \vec{\epsilon}(t) dt \tag{A.47}$$

$$\text{Size N representation} = \text{True representation} - \text{Error} \tag{A.48}$$

The last term is indeed the second type error that we have discussed in the error analysis section of the paper. Using eigenvalue decomposition of $A = V\Lambda V^{-1}$ we re-write the above error term as

$$\text{Error} = V \int_{t=0}^{T} \frac{1}{t} \exp\left(\Lambda \ln \frac{t}{T}\right) V^{-1} \vec{\epsilon}(t) dt \tag{A.49}$$

with a change of variable $s = \ln \frac{t}{T}$

$$V^{-1} \text{Error} = \int_{t=0}^{T} \exp(\Lambda s) V^{-1} \vec{\epsilon}(s) ds \tag{A.50}$$

According to the assumption of this theorem $\|V^{-1} \epsilon(t)\|_2 = \|\epsilon(t)\|_2 \leq \epsilon_m$

$$\to [V^{-1} \text{Error}]_j \leq \int_{t=0}^{T} \exp(s\lambda_j) \epsilon_m ds = \frac{\epsilon_m}{\lambda_j} \tag{A.51}$$

$$\|\text{Error}\|^2 = \|V^{-1} \text{Error}\|^2 \leq \epsilon_m \sqrt{\sum \frac{1}{\lambda_i^2}} = \epsilon_m \|A^{-1}\|_F \tag{A.52}$$

**Theorem** (5). *Suppose a frame $\Phi$ is given. The Dual of Truncation (DoT) construct introduced in Section 5.1.3 has optimal representation error when compared to any other SaFARi$^{(N)}$ construct for the same frame $\Phi$.*

**Proof**: The proof for this theorem involves two steps.

1. First, we show that the optimal representation error in the theorem can be reduced to optimal error of the second type (mixing).

2. Then, we demonstrate that for a given frame $\Phi$, and given truncation level $N$, the construct with the optimal second type error control is DoT.

As discussed in Sec. 5.2, the first type of error is due to truncating the frame, and is independent of the SSM. In the scope of this theorem, all the SaFARi$^{(N)}$ constructs use the same frame and the same truncation. Therefore, comparing the representation error between SaFARi$^{(N)}$ in the theorem reduces to comparing the mixing error.

The mixing error is shown in Theorem 3 to be proportional to

$$\vec{\epsilon}(T) = \langle u_T, \xi \rangle \tag{A.53}$$

To minimize $\|\vec{\epsilon}(T)\|$ irrespective of the input signal, one has to minimize $\|\xi\|_F^2$

$$\xi = \Upsilon_{[0:N]}(\widetilde{\Phi}^{(N)}\Phi_{[0:N]} - I) \tag{A.54}$$

Where $\Upsilon_{[0:N]}$ and $\Phi_{[0:N]}$ are the first $N$ elements of $\Upsilon$ and $\Phi$. $\widetilde{\Phi}^{(N)}$ is the approximation of the dual frame that determines the SaFARi$^{(N)}$ construction. For the ease of notation, we rewrite Eq. A.54 as:

$$\xi = \Upsilon(\widetilde{\Phi}\Phi - I) \tag{A.55}$$

For a fixed $\Upsilon$, and $\Phi$, we aim to minimize $\|\xi\|_F^2$ with respect to $\widetilde{\Phi}$:

$$Argmin_{\widetilde{\Phi}}\|\Upsilon(\widetilde{\Phi}\Phi - I)\|_F^2 \tag{A.56}$$

For the optimal $\widetilde{\Phi}$, the partial derivative is zero.

$$\frac{\partial}{\partial\widetilde{\Phi}}\|\xi\|_F^2 = \frac{\partial}{\partial\widetilde{\Phi}}\|\Upsilon(\widetilde{\Phi}\Phi - I)\|_F^2 = 2\Upsilon\Upsilon^T(\widetilde{\Phi}\Phi - I)\Phi^T = 0 \tag{A.57}$$

$$\widetilde{\Phi} = (\Phi\Phi^T)^{-1}\Phi^T \tag{A.58}$$

One should note that the described $\widetilde{\Phi}$ is indeed the pseudo-inverse dual for the truncated frame $\Phi_{[0:N]}$. Therefore, among the possible SaFARi$^{(N)}$ constructs for the same frame, the Dual of Truncation (DoT) has optimal representation error. $\square$

### A.7 Parallelization using the convolution kernel

**Theorem** (6). *For SaFARi using the scaling measure, if $A$ is diagonalizable, computing the scaled representation on a sequence with $L$ samples can be done using a kernel multiplication.*
*a) For the discretization using General Biliniear Transform (GBT) with parameter $\alpha$, the kernel can be computed using:*

$$K_L[i,j] = \frac{\prod_{k=j+1}^{L}\left(1 - \frac{1-\alpha}{k+1}\lambda_i\right)}{\prod_{k=j}^{L}\left(1 + \frac{\alpha}{k+1}\lambda_i\right)} \in \mathbb{R}^{N\times L} \tag{A.59}$$

*b) For long sequences, the kernel $K$ can be approximated using*

$$K_L(i,j) = \frac{1}{j}\left(\frac{j}{L}\right)^{\lambda_i} \in \mathbb{R}^{N\times L} \tag{A.60}$$

*For either case a or b, the representation is computed as:*

$$c = MK_L\vec{u}, \quad M = V\text{diag}(V^{-1}B) \tag{A.61}$$

*where $V$ and $\lambda_i$ are the eigenvector matrix and eigenvalues of $A$.*

**Proof**: a) rewriting the GBT update rule for the diagnoalzied SSM we have:

$$\widetilde{C}[n+1] = \left(I + \frac{\alpha}{n+1}\Lambda\right)^{-1}\left(I - \frac{1-\alpha}{n+1}\Lambda\right)\widetilde{C}[n] + \left(I + \frac{\alpha}{n+1}\Lambda\right)^{-1}\widetilde{B}u[n] \tag{A.62}$$

$$= \bar{A}_n\widetilde{C}[n] + \bar{B}_nu[n]$$

$$\rightarrow \widetilde{C}[L+1] = \bar{B}_Lu[L] + \bar{A}_L\bar{B}_{L-1}u[L-1] + ... + \bar{A}_L...\bar{A}_1\bar{B}_0u[0] \tag{A.63}$$

for the ease of computation and notation, we define $K_L$ such that:

$$K_L[i,j] = \bar{A}_L...\bar{A}_{j+1}\left(1 + \frac{\alpha}{j+1}\lambda_i\right)^{-1} = \frac{\prod_{k=j+1}^L\left(1 - \frac{1-\alpha}{k+1}\lambda_i\right)}{\prod_{k=j}^L\left(1 + \frac{\alpha}{k+1}\lambda_i\right)} \tag{A.64}$$

$$\widetilde{C}_i = \widetilde{B}_i\sum_j K_L[i,j]u[j] = \widetilde{B}_i[K_L\vec{u}]_i \tag{A.65}$$

$$\widetilde{c} = \widetilde{B}\odot(K_L\vec{u}) = \mathrm{diag}(\widetilde{B})K_L\tilde{u} \tag{A.66}$$

$$c = V\widetilde{c} = V\mathrm{diag}(\widetilde{B})K_L\tilde{u} = MK_L\tilde{u} \tag{A.67}$$

**Proof**: b) Using Lemma 1 for the diagonalized version of Scaled-SaFARi the closed form solution is

$$\widetilde{c} = \int_{t=0}^T \frac{1}{t}\exp\left(\Lambda\ln\frac{t}{T}\right)\widetilde{B}u(t)dt. \tag{A.68}$$

$$\widetilde{c}_i = \int_{t=0}^T \frac{1}{t}\exp\left(\lambda_i\ln\frac{t}{T}\right)\widetilde{B}_iu(t)dt = \widetilde{B}_i\int_{t=0}^T \frac{1}{t}\left(\frac{t}{T}\right)^{\lambda_i}u(t)dt = \widetilde{B}_i[K_L\vec{u}]_i \tag{A.69}$$

$$\widetilde{c} = \widetilde{B}\odot(K_L\vec{u}) = \mathrm{diag}(\widetilde{B})K_L\tilde{u} \tag{A.70}$$

$$c = V\widetilde{c} = V\mathrm{diag}(\widetilde{B})K_L\tilde{u} = MK_L\tilde{u} \tag{A.71}$$

**Theorem** (7). *For SaFARi using translated measure with $\theta_L$ samples long sliding window,if A is diagonalizable, computing the translated representation on a sequence with L samples can be done using a kernel multiplication.*

*a) for the discretization using General Bilinear Transform(GBT) with parameter $\alpha$, the kernel can be computed using:*

$$K_L[i,j] = \frac{1}{1 + \frac{\alpha}{\theta}\lambda_i}\left(\frac{1 - \frac{1-\alpha}{\theta}\lambda_i}{1 + \frac{\alpha}{\theta}\lambda_i}\right)^{L-j} \tag{A.72}$$

*b) For long sequences, the kernel K can be approximated using*

$$K_L(i,j) = \frac{1}{\theta_L}\exp\left(-\lambda_i\frac{L-j}{\theta_L}\right) \in \mathbb{R}^{N\times L} \tag{A.73}$$

*For either of case a or b, the representation is computed by*

$$c = MK_L\vec{u}, \quad M = V\mathrm{diag}(V^{-1}B). \tag{A.74}$$

*where V and $\lambda_i$ are the eigenvectors matrix, and eigenvalues of A.*

**Proof**: a) rewriting the GBT update rule for the diagonalized SSM we have:

$$\widetilde{C}[n+1] = \left(I + \frac{\alpha}{\theta}\Lambda\right)^{-1}\left(I - \frac{1-\alpha}{\theta}\Lambda\right)\widetilde{C}[n] + \left(I + \frac{\alpha}{\theta}\Lambda\right)^{-1}\widetilde{B}u[n] = \bar{A}\widetilde{C}[n] + \bar{B}u[n] \tag{A.75}$$

we take a similar approach to the previous theorem. The only difference is that $\bar{A}$ and $\bar{B}$ remain the same for all the time indices.

$$\widetilde{C}[L+1] = \bar{B}u[L] + \bar{A}\bar{B}u[L-1] + \dots + \bar{A}^L\bar{B}u[0] \tag{A.76}$$

for the ease of computation and notation, we define $K_L$ such that:

$$K_L[i,j] = \frac{1}{1 + \frac{\alpha}{\theta}\lambda_i}\left(\frac{1 - \frac{1-\alpha}{\theta}\lambda_i}{1 + \frac{\alpha}{\theta}\lambda_i}\right)^{L-j} \tag{A.77}$$

$$\widetilde{C}_i = \widetilde{B}_i \sum_j K_L[i,j]u[j] = \widetilde{B}_i[K_L\vec{u}]_i \tag{A.78}$$

$$\widetilde{c} = \widetilde{B} \odot (K_L\vec{u}) = \mathrm{diag}(\widetilde{\mathrm{B}})\mathrm{K_L}\tilde{\mathrm{u}} \tag{A.79}$$

$$c = V\widetilde{c} = V\mathrm{diag}(\widetilde{\mathrm{B}})\mathrm{K_L}\tilde{\mathrm{u}} = \mathrm{MK_L}\tilde{\mathrm{u}} \tag{A.80}$$

**Proof**: b) Using Lemma 2 for the diagonalized version of Translated-SaFARi, the closed-form solution is:

$$\widetilde{c} = \int_{t=T-\theta}^{T} \frac{1}{\theta}\exp\left(\Lambda\frac{t-T}{\theta}\right)\widetilde{B}u(t). \tag{A.81}$$

$$\widetilde{c}_i = \int_{t=T-\theta}^{T} \frac{1}{\theta}\exp\left(\lambda_i\frac{t-T}{\theta}\right)\widetilde{B}_iu(t) = \widetilde{B}_i[K_L\vec{u}]_i \tag{A.82}$$

$$\widetilde{c} = \widetilde{B} \odot (K_L\vec{u}) = \mathrm{diag}(\widetilde{\mathrm{B}})\mathrm{K_L}\tilde{\mathrm{u}} \tag{A.83}$$

$$c = V\widetilde{c} = V\mathrm{diag}(\widetilde{\mathrm{B}})\mathrm{K_L}\tilde{\mathrm{u}} = \mathrm{MK_L}\tilde{\mathrm{u}} \tag{A.84}$$

### A.7.1 Numerical instability of the fast sequential LegS solver

As part of our experimental findings, we realized that the proposed method for sequential updates for LegS SSM( in Gu et al. (2020),Appendix E) suffers from numerical instability when working with larger SSMs.

$$x = \frac{\mathrm{cumsum}(\beta\mathrm{cumprod}\frac{1}{\alpha})}{\mathrm{cumprod}\frac{1}{\alpha}} \tag{A.85}$$

where the introduced $\alpha_k = \frac{d_k}{1+d_k}$ is a decreasing function. One can confirm that in the $t^{\mathrm{th}}$ iteration, and for the $k^{\mathrm{th}}$ degree Legendre polynomial

$$\alpha_k = \frac{d_k}{1+d_k} = \frac{2(t+1)-k}{2(t+1)+k+1} \tag{A.86}$$

Then the proposed solution requires finding cumulative product of $\frac{1}{\alpha_k}$ for $k \in [1, N]$ in each step.

$$\log\left|\mathrm{cumprod}_{k'}\left(\frac{1}{\alpha_{k'}}\right)(k)\right| = \sum_{k'=1}^{K}\log\left|\frac{1}{\alpha_{k'}}\right| = \sum_{k'=1}^{K}\log\left|-1 + \frac{4(t+1)+1}{2(t+1)-k'}\right| \tag{A.87}$$

$$= \sum_{k'=1}^{2(t+1)}\log\left(-1 + \frac{4(t+1)+1}{2(t+1)-k'}\right) + \sum_{k'=2(t+1)+1}^{K}\log\left(1 + \frac{4(t+1)+1}{k'-2(t+1)}\right) \tag{A.88}$$

$$= \sum_{k'=1}^{2(t+1)}\log\left(-1 + \frac{4(t+1)+1}{2(t+1)-k'}\right) + \sum_{k'=1}^{K-2(t+1)}\log\left(1 + \frac{4(t+1)+1}{k'}\right) \tag{A.89}$$

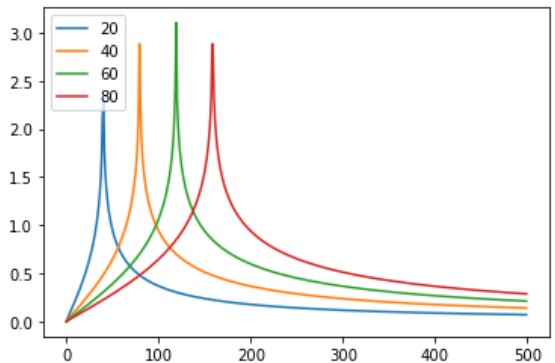 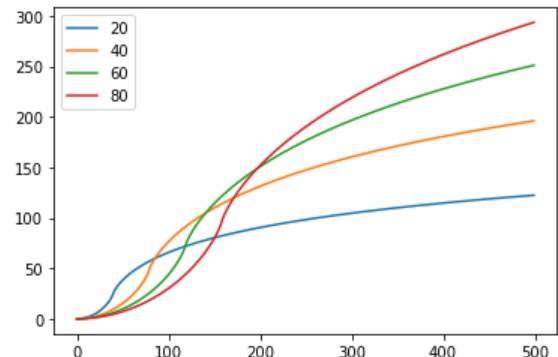

Figure 10: Fast Legs numerically diverges. **left:** For a system with $N = 500$, $\log|\frac{1}{\alpha_k}|$ for different values of $k \in [1, N]$ is plotted at different iterations $t = 20, 40, 60, 80$. **right:** log of the cumulative product which is equal to the cumulative sum of the left plot is plotted for different iterations. In the right plot, it is notable that for $t = 80$, the cumulative product reaches to $10^{300}$ for a $k < 500$ which is the largest value that a float-64 variable can handle. The studied Fast-LegS method for an SSM having more then 500 coefficients diverges after only 80 sequential updates.

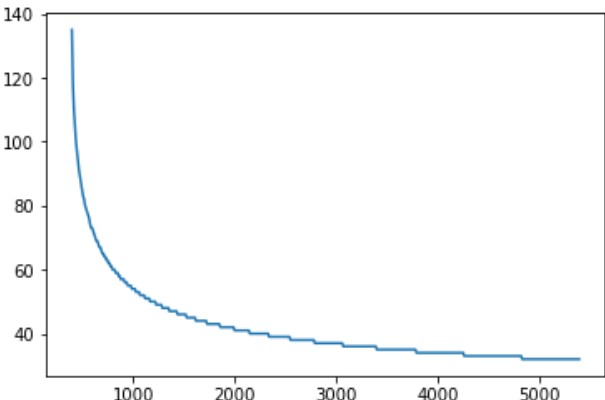

Figure 11: As the given LegS size $N$ grows, the longest sequence length before observing numerical diversion is given in the above plot. For $N < 400$ we did not observe the numerical diversion. for $N > 400$ , the fast legS sequential update method cannot handle sequences longer than a limited length before it becomes numerically unstable.

For any specific iteration (fixed $t$), as K grows(higher representation index), the second summation above grows to infinity. Thus, for large enough N, cumprod($\frac{1}{\alpha_k}$) diverges beyond machine precision. As a result, the proposed fast sequential legS solver proposed in (Gu et al. (2020), Appendix E) fails. Figure. 10 Shows an example where for $N = 500$, fast LegS numerically diverges for any sequence longer than 80 samples. It is crucial to note that this numerical instability is fundamental to legS, and does not depend on the input signal at all.

We also investigate the longest sequence length that the given fast legS sequential solution can handle without numerical diversion. Figure 11 shows that as $N$ grows, then length of sequence that fast legS sequential solver can handle before becoming numerically unstable decrease to a limited length.

A stable version of solving LegS would be a similar approach as fast-LegS, but in the last step, instead of introducing the proposed $\alpha$ and $\beta$, we find $x_1$, then recursively find $x_i$ after finding all the pervious $x_i$s. This way, the overall computation complexity remains the same, while the run-time complexity increases, as one cannot compute $x_i$ until $x_0, \ldots, x_{i-1}$ computed.

## A.8 Bases and frames

**Legendre Polynomials:** Legendre polynomials form an orthonormal basis for square-integrable functions on the compact interval $[-1, 1]$. Their recurrence relations and bounded values make them well-suited for tasks such as Gaussian quadrature, spectral methods for solving differential equations, and polynomial interpolation. However, SSMs built from Legendre polynomials can suffer from numerical instability for high degrees, require global support (making them less effective for localized features), and may produce oscillations near interval boundaries, limiting their practicality in approximating functions with sharp variations or discontinuities.

**Fourier Series:** The familiar Fourier series is an orthonormal basis, and with integer $n$ elements have the form: $\sqrt{2}\cos(2\pi nt)$, $\sqrt{2}\sin(2\pi nt)$. The Fourier series represents functions as infinite sums of sines and cosines, which form an orthonormal basis for square-integrable functions on a compact interval, making them highly effective for approximating periodic functions. However, they have important limitations for function approximation: they assume periodicity, and can exhibit the Gibbs phenomenon when approximating non-periodic or sharply varying functions; they provide global support, making them inefficient for capturing localized features; and high-frequency components are often needed to approximate functions with sharp transitions, which can be computationally expensive.

**Random Harmonics:** Similar to the Fourier series, these functions have the form $\sqrt{2}\cos(2\pi at)$, $\sqrt{2}\sin(2\pi at)$, where $a$ is a real number sampled from a uniform distribution, not integers. A random collection of these functions is not guaranteed to span L2, and therefore is not a frame or basis. In general, using random vectors in place of a frame or basis is a poor choice. We include it separately only as a counter-example to orthogonal bases and redundant frames. We can also use this set to augment a Fourier basis: since a Fourier basis spans L2, concatenating additional vectors introduces redundancy, and we have a frame.

**Laguerre Polynomials:** Laguerre polynomials form an orthonormal basis for square-integrable functions on the semi-infinite interval $[0, \infty]$ with respect to the weight function $e^{-x}$. These properties make them well-suited for approximating functions defined on unbounded domains, particularly when the function decays exponentially, and they are widely used in quantum mechanics, numerical analysis, and differential equations. Laguerre polynomials are less effective for approximating functions on compact intervals, as they exhibit increasingly oscillatory behavior for large degrees, and their reliance on a specific weight function limits their flexibility for approximating functions that do not exhibit exponential decay.

**Chebyshev Polynomials:** Chebyshev polynomials form an orthonormal (or orthogonal, depending on normalization) basis for square-integrable functions on the compact interval $[-1, 1]$ with respect to the weight $\frac{1}{\sqrt{1-x^2}}$. They are known for their near-minimax property, meaning polynomial approximations using Chebyshev nodes minimize the maximum error. This makes them highly effective in polynomial interpolation, spectral methods for solving differential equations, and numerical approximation schemes that need high accuracy with fewer terms. However, Chebyshev polynomials are global basis functions, so they are less effective for functions with localized features or sharp discontinuities. They also require transformations or rescaling for domains other than $[-1, 1]$, and like other polynomial bases, they can be inefficient for very high-dimensional or highly irregular function approximations.

**Bernstein Polynomials:** Bernstein polynomials form a basis for continuous functions on the compact interval $[0, 1]$. They are non-negative, form a partition of unity, and have excellent shape-preserving properties, making them particularly useful in computer graphics, geometric modeling (e.g., Bézier curves), and approximation theory. Bernstein polynomials are not orthogonal, which limits their efficiency in some numerical computations, and achieving high accuracy often requires large polynomial degrees, leading to higher computational cost. They are also less suitable for capturing oscillatory behavior or sharp transitions, as compared to orthogonal polynomial bases like Chebyshev or Legendre.

**Gabor:** Gabor filters (or Morlets) capture localization in both time and frequency, making them popular for use in biomedical signal processing and image texture analysis. They can be constructed by modulating a complex sinusoid with a Gaussian as: $f_k(x) = e^{-(x-b_k)^2/w^2} e^{-ia_k(x-b_k)}$, where $b_k$ are shifts in time, $a_k$ are frequencies, and $w$ is the scaling of a Gaussian envelope. To ensure that the resulting set of functions creates an oversampled lattice, we ensure that the time-frequency sampling overlaps; that is, $\Delta a \Delta b < 1$.

### A.9 Additional Experimental Results

Tables 4 and 5 provide the standard deviation for the MSE results in Table 3. Most values are on the same order of magnitude. A notable exception is "Rand". Since the collection of random sines and cosines do not form a complete frame or basis, it is a matter of luck how well they represent a particular signal. The more random components are added, the more likely it is that the vectors will support the signal of interest, and shorter signals require fewer components. Since the translated measure has a consistently small window, the variance here is on par with other true bases/frames. The worst case is when the signal is long (scaled measure) and the rank is low (N=32), as expected.

| | Scaled | | | | | |
| | $N = 32$ | | $N = 64$ | | $N = 128$ | |
| | SP | M4 | SP | M4 | SP | M4 |
|---|---|---|---|---|---|---|
| Leg | 0.0007 | 0.0062 | 0.0007 | 0.0060 | 0.0007 | 0.0060 |
| Fou | 0.0003 | 0.0040 | 0.0002 | 0.0029 | 0.0002 | 0.0020 |
| Lag | 0.0012 | 0.0100 | 0.0012 | 0.0100 | 0.0012 | 0.0100 |
| Cheby | 0.0003 | 0.0030 | 0.0003 | 0.0021 | 0.0003 | 0.0014 |
| Bern | 0.0005 | 0.0037 | 0.0004 | 0.0032 | 0.0003 | 0.0029 |
| Rand | 0.0005 | 0.0258 | 0.0004 | 0.0105 | 0.0003 | 0.0048 |
| Fou+ | 0.0003 | 0.0036 | 0.0002 | 0.0028 | 0.0002 | 0.0019 |
| Fou+* | 0.0002 | 0.0029 | 0.0002 | 0.0018 | 0.0001 | 0.0010 |
| Gabor | 0.0004 | 0.0050 | 0.0003 | 0.0033 | 0.0125 | 0.0040 |
| Gabor* | 0.0003 | 0.0041 | 0.0003 | 0.0035 | 0.0009 | 0.0026 |

Table 4: Standard deviation of reconstructed signals with different instantiations of SaFARi and a scaled measure. The table is divided into polynomial (top) and non-polynomial (bottom) representations.

| | Translated | | | | | |
| | $N = 32$ | | $N = 64$ | | $N = 128$ | |
| | SP | M4 | SP | M4 | SP | M4 |
|---|---|---|---|---|---|---|
| Leg | 0.0029 | 0.0069 | 0.0029 | 0.0068 | 0.0029 | 0.0069 |
| Fou | 0.0006 | 0.0052 | 0.0006 | 0.0051 | 0.0006 | 0.0052 |
| Lag | 0.0006 | 0.0052 | 0.0006 | 0.0051 | – | – |
| Cheby | 0.0006 | 0.0052 | 0.0008 | 0.0052 | 0.0011 | 0.0054 |
| Bern | 0.0006 | 0.0052 | 0.0006 | 0.0051 | 0.0006 | 0.0052 |
| Rand | 0.0006 | 0.0053 | 0.0007 | 0.0052 | 0.0009 | 0.0053 |
| Fou+ | 0.0006 | 0.0052 | 0.0007 | 0.0051 | 0.0009 | 0.0053 |
| Fou+* | 0.0006 | 0.0053 | 0.0008 | 0.0052 | 0.0009 | 0.0054 |
| Gabor | 0.0006 | 0.0052 | 0.0006 | 0.0051 | 0.0007 | 0.0053 |
| Gabor* | 0.0006 | 0.0053 | 0.0007 | 0.0051 | 0.0007 | 0.0053 |

Table 5: Standard deviation of reconstructed signals with different instantiations of SaFARi and a translated measure. The table is divided into polynomial (top) and non-polynomial (bottom) representations. Missing entries for Laguerre could not be computed due to numerical errors arising from exponents in higher-order terms.

