# OpenReview forum: "SaFARi: State-Space Models for Frame-Agnostic Representation"
_TMLR — Accepted by TMLR_

### Review · Reviewer_8VQc · 2025-08-01

**Summary Of Contributions:**

In this paper, the authors derive general equations for State Space Models (SSM) where any frame may be used for function approximation. This generalizes earlier results mostly based on various orthogonal polynomials (so-called HiPPO method). Instead of closed-form expressions, they obtain SSMs were the update operators are under the form of various inner product between frames and their duals. They then discuss quite exhaustively matters of finite approximation, resulting errors, and computational complexity. The appendices are quite extensive.

**Audience:**

Yes

**Claims And Evidence:**

Yes

**Requested Changes:**

See above.

**Strengths And Weaknesses:**

Strengths:
- a novel (to my knowledge) but natural derivation
- a quite extensive exposition, with interesting discussions on practical implementation, relation to prior work, and so on

Weaknesses:
- there are no numerical experiments (or almost). I understand that this is a "resubmitted" shorter version that focuses on the theoretical aspects (I do not have access to the previous version), however it is difficult to visualize and understand how the proposed method "generalize" previous works to "any frame" without providing one single new example and demonstrating its numerical performance and its implementation, its error bound, and so on. Is there a particularly striking new example of frames that is significantly different from previous work, and whose implementation can be done with the results of the paper?
- it does not seem that $\Phi$ being a frame (or even a basis) plays any role in the theoretical analysis? If not, is it possible to indicate why it is important to have a frame? (beyond obvious numerical performance)
- The section on truncating and error bound is a bit unrigorous. It is not immediate to define the convergence of growing matrices towards operators (def. 4): one must introduce proper norms, hypotheses of summability, etc. Following that, subsequent claims that truncations "obviously converge" must be handled a bit more carefully.
- Related to the two previous points, one would expect to see the $A_{frame}, B_{frame}$ appear somewhere in the bounds? If not, why?

---

> ### Author Response · Authors · 2025-09-13
> **Response to reviewer 8VQc**
>
> Dear Reviewer,
>
> We thank you for your detailed and thoughtful feedback.  We have addressed your points below, and provided responses here as well as edits and updates to our manuscript as appropriate.  Please note the blue text in our updated manuscript, which indicates additions since the last submission.
>
> **There are no numerical experiments (or almost).**
>
> While this paper was intended to be theoretical in scope, we agree that it would benefit from quantitative examples.  To this end, we added a new section to the paper entitled Empirical Evidence. In this section, we use the SaFARi method to instantiate SSMs on different frames and bases.  Several of these orthogonal polynomial bases were studied in prior work, and we include them for two reasons: as baselines for comparison, and to demonstrate that our method indeed replicates prior work while generalizing beyond it.  We also instantiate SSMs on a non-orthogonal basis, two frames, and a collection of vectors that forms neither a basis nor a frame.
>
> In one experiment, we use our SaFARi-based SSMs to sequentially process long time-series data (from the M4 and S&P 500 datasets) and compare their resulting MSEs.  Then, to compare their performance to learned models, we provide results of a delay task, where models reconstruct a delayed copy of the signal.  Table 3 and 4 are in progress; the pending results will not alter the key takeaway that a chosen frame/measure should be task-specific, and SaFARi provides that flexibility.
>
> Additionally, we make our code available for the reader: the link is provided with the paper, and will be changed from anonymous to a publicly available Github repo after the review period.
>
> **…it does not seem that  being a frame (or even a basis) plays any role in the theoretical analysis.**
>
> The frame condition must be met for two reasons: to adequately represent a signal, and to construct the SSM.  In our original paper, we used this assumption implicitly in several locations, but in order to make this connection explicit for our readers, we have included the following language to Sec. 4.4:
>
> *"While the numerical method described here could be applied to any differentiable set of vectors, we require that the vectors form a frame.  If not, then projecting the input signal onto the given vectors is lossy and not invertible.  More precisely, the frame condition (Eq. 7) is necessary and sufficient for the frame operator to be invertible on its image with a bounded inverse. This makes the frame a complete and stable (though potentially redundant) framework for signal representation.  Second, and most importantly for this work, if $\phi$ does not meet the frame condition, then Eq. 9 does not hold, and the derivative of representation with respect to time cannot be calculated using only the current hidden state."*
>
> To further support this point, we include the collection of random harmonics in our experimental results, and it is clear that compared to frames and bases, the performance is quite poor.
>
> **The section on truncating and error bound is a bit unrigorous. It is not immediate to define the convergence of growing matrices towards operators...  one would expect to see the Aframe, Bframe appear somewhere in the bounds? If not, why?**
>
> This is a keen observation.  Although the error bound does not explicitly have $A, B$ in its formula, one can rewrite the error bound as:
> $$\xi = \Upsilon ( \widetilde{\Phi} \Phi - I ) \rightarrow \xi =  \Upsilon \widetilde{\Phi} \Phi - \Upsilon$$
> Since $A=I+  \Upsilon \widetilde{\Phi}$ we can rewrite the above as: $$ \xi =(A-I) \Phi - \Upsilon$$
> which reflects that the error bound indeed depends on $A$. The dependence on $B$ is not readily obvious; rather than explicitly depending on ${\phi(t=1)}$, the error bound depends on the whole of $\Phi$.
>
> Regarding convergence, we have added text to Section 5.1.1 to describe the type of convergence we expect, and have taken care to explain that we make this assumption for the following sections in order to be able to compare different constructions, without making specific claims.  Our implementations support that the sequences do indeed converge in practice; however, we are still short of a rigorous proof, and we acknowledge this theoretical gap in the detailed explanation in the Appendix.  We anticipate addressing this question in follow-up work, and would welcome input from interested members of the community as we continue to work towards a more complete answer to this question.
>
> We wish to thank the reviewer once more for their thoughtful feedback that has led to significant improvements in the content and clarity of our paper.

---

> > ### Author Response · Authors · 2025-10-12
> > **Update**
> >
> > Dear Reviewer,
> >
> > We are finalizing additional experiments and responses as requested, and expect to have an updated manuscript with completed experimental results within the week.  We look forward to sharing it with you shortly.

---

> > > ### Author Response · Authors · 2025-10-17
> > > **Updated manuscript**
> > >
> > > Dear Reviewer,
> > >
> > > As promised in our previous comment, we have provided you an updated version of our manuscript with complete experimental results.  The note accompanying our updated manuscript describes the changes since the last version, which are as follows:
> > >
> > > 1) The planned experimental results in Section 7 are complete, and additional discussion has been added where appropriate.
> > > 2) Section 7.1.2 (function approximation) now includes a discussion of rank vs. effective rank for SSMs constructed from redundant frames.  As our paper advocates for moving beyond polynomial bases to handle cases where closed-form solutions do not exist, addressing redundancy and rank deficiency adds important context. Two additional entries in Table 3 are included for comparison of full-rank and rank-deficient SSMs in the experimental section.
> > > 3) Table 4 has been replaced by Figure 9.  Section 7.1.3 (the delay/copying experiment) has been extended to include a much more densely sampled set of delays, which were better represented as a plot than a table.
> > > 4) A summary of key takeaways has been added to the conclusion, which we hope will clarify and crystallize the main messages of our paper for readers.
> > > 5) Miscellaneous edits for clarity have been made as recommended by Reviewer LaW1.
> > >
> > > We thank you again for your thoughtful feedback during the review process, and we will be happy to provide any further information you request.

---

### Review · Reviewer_QyCH · 2025-08-03

**Summary Of Contributions:**

The work proposed SaFARi (SSMs for Frame-Agnostic Representation), a generalizable method for building an SSM with any basis and framework and extends existing methods like HiPPO which are restricted to polynomial basis. To address the challenge that closed-form solution is not available, the work provides numerical methods which apply to both uniform scaled and uniform translated measures. The work shows theoretical study results on the error bound of the approach. The work shows the proposed approach can run in time complexity as low as O(N Log L), demonstrating superior computational efficiency against existing methods.

**Audience:**

Yes

**Broader Impact Concerns:**

As a work primarily focusing on methodology and results, I don't have big concerns for its broader impacts.

**Claims And Evidence:**

Yes

**Requested Changes:**

I would like to see the work to include some quantitative study results and make comparisons against existing SSMs like HiPPO or learning-based methods like LSTM or transformers.

**Strengths And Weaknesses:**

**Strengths**
1. A major strength of the work is the generalizable applicability to any frame or basis. In contrast, previous approach makes assumptions about the basis. For example, HiPPO requires a polynomial basis.
2. The work include very solid theoretical study on the error bounds of the work. Even though the proposed framework is based on numerical methods and subject to numerical errors, the theoretical study results provides a comprehensive and in-depth analysis on this error which provides better guarantee for the applicability of the proposed framework.
3. The work provides technical details on the implementation and computational efficiency analysis.

** Weakness**
A major weakness of the work is lack of quantitative results. There're many tasks the proposed approach can apply to like forecasting or inference. Many existing SSMs or deep-learning based methods like LSTM or transformers can be applied to these tasks as well. The work should compare their methods against these methods on these tasks and provide quantitative results.

---

> ### Author Response · Authors · 2025-09-13
> **Response to reviewer QyCH**
>
> Dear Reviewer,
>
> We thank you for your thoughtful feedback.  We have addressed your points below, and provided responses here as well as edits and updates to our manuscript as appropriate.
>
> **I would like to see the work to include some quantitative study results and make comparisons against existing SSMs like HiPPO or learning-based methods like LSTM or transformers.**
>
> While our initial submission was intended as a purely theoretical contribution, we agree that including quantitative experiments significantly strengthens the work.  Therefore, we have added a new section (7. Empirical validation), which demonstrates how our method can be used to instantiate SSMs–including the orthogonal polynomials of prior work, as well as redundant frames, non-orthogonal polynomials, and even collections of random vectors.  We also compare our SSMs with popular learning-based models such as LSTMs and GRUs.  Table 3 and 4 are in progress; the pending results will not alter the key takeaway that a chosen frame/measure should be task-specific, and SaFARi provides that flexibility.
>
> This section is not intended as an ablation study.  While this paper remains focused on establishing the theoretical foundations and principles of SaFARi, we view this experimental section as an important step in illustrating its power and flexibility. We anticipate that future research, including work by the broader community, will further explore which frames achieve optimal performance for specific domains.
>
> As we hope to disseminate this work more broadly, we are also providing code with our paper (anonymously during the review period) that allows a user to effortlessly generate SSMs with built-in or custom frames.
>
> **... is it possible to indicate why it is important to have a frame? (beyond obvious numerical performance)**
>
> The frame condition must be met for two reasons: to adequately represent a signal, and to construct the SSM.  In our original paper, we used this assumption implicitly in several locations, but in order to make this connection explicit for our readers, we have included the following language to Sec. 4.4:
>
> *"While the numerical method described here could be applied to any differentiable set of vectors, we require that the vectors form a frame.  If not, then projecting the input signal onto the given vectors is lossy and not invertible.  More precisely, the frame condition (Eq. 7) is necessary and sufficient for the frame operator to be invertible on its image with a bounded inverse. This makes the frame a complete and stable (though potentially redundant) framework for signal representation.  Second, and most importantly for this work, if $\phi$ does not meet the frame condition, then Eq. 9 does not hold, and the derivative of representation with respect to time cannot be calculated using only the current hidden state."*
>
> We thank the reviewer again for their time and insightful comments, and the opportunity to improve our paper.

---

> > ### Author Response · Authors · 2025-10-12
> > **Update**
> >
> > Dear Reviewer,
> >
> > We are finalizing additional experiments and responses as requested, and expect to have an updated manuscript with completed experimental results within the week.  We look forward to sharing it with you shortly.

---

> > > ### Author Response · Authors · 2025-10-17
> > > **Updated manuscript**
> > >
> > > Dear Reviewer,
> > >
> > > As promised in our previous comment, we have provided you an updated version of our manuscript with complete experimental results.  The note accompanying our updated manuscript describes the changes since the last version, which are as follows:
> > >
> > > 1) The planned experimental results in Section 7 are complete, and additional discussion has been added where appropriate.
> > > 2) Section 7.1.2 (function approximation) now includes a discussion of rank vs. effective rank for SSMs constructed from redundant frames.  As our paper advocates for moving beyond polynomial bases to handle cases where closed-form solutions do not exist, addressing redundancy and rank deficiency adds important context. Two additional entries in Table 3 are included for comparison of full-rank and rank-deficient SSMs in the experimental section.
> > > 3) Table 4 has been replaced by Figure 9.  Section 7.1.3 (the delay/copying experiment) has been extended to include a much more densely sampled set of delays, which were better represented as a plot than a table.
> > > 4) A summary of key takeaways has been added to the conclusion, which we hope will clarify and crystallize the main messages of our paper for readers.
> > > 5) Miscellaneous edits for clarity have been made as recommended by Reviewer LaW1.
> > >
> > > We thank you again for your thoughtful feedback during the review process, and we will be happy to provide any further information you request.

---

### Review · Reviewer_LAw1 · 2025-08-16

**Summary Of Contributions:**

The paper introduces a method for computing transition matrices in state spaced models (SSMs) for arbitrary frames, extending the popular HiPPO approach for SSMs.

In theory, this new method improves on multiple weaknesses of HiPPO, namely the non-diagonalizability of the ubiquitous scaled Legendre (LegS), by choosing other frames and measures that should result in more stable dynamics. In this context, the submission also introduces the notion of truncation and mixing errors, and shows how to optimally construct the dual with finite-dimensional approximations.

**Audience:**

Yes

**Claims And Evidence:**

Yes

**Requested Changes:**

Requested changes:

1. Empirical validation: For example replication of experiments from HiPPO (pMNIST etc). The validation should show that the matrices derived with the presented method work in practice, at least on toy problems, and that in some scenarios not commonly used frames are superior to LegS.

2. The claim in Section 2.3 – that many of the bases perform quite poorly – is rather strong and requires more citations.

Additional questions:

1. Why are only uniformly weighted measures considered ?

2. How does this method benefit large-scale SSMs?

3. What is the usecase of more general frames, especially non-orthogonal basis functions? How do these influence performance of the algorithm?

4. The explanation why A.3 has different results from HiPPO is not good enough – the authors should check if indeed they get to the same results when using a different Fourier basis or range, and otherwise reach out to HiPPO authors trying to understand where the difference comes from.

5. Does SaFARi allow for the design of new gating mechanisms in RNNs?

6. Is Theorem 4 really a tight bound? The Frobenius norm could be quite large.

Additional comments:

1. Diagonalizing the transition matrix A in section 2.2: This section has multiple imprecise statements that only hold under certain conditions, in particular the authors neglect the case of having degenerate eigenvalues and subspaces of eigenvectors. Should this be mentioned somewhere?

2. Bad line break at end of page 2.

3. Using variable "x" in Equations (8ff) is somewhat confusing.

4. The variable mu in Equation (12) should be explicitly introduced.

5. Text just before (A.12) should probably say "We can write the coefficients as"

**Strengths And Weaknesses:**

Strengths:

1. The method to compute transition matrices for SSMs with any frame or basis seems novel.

2. Introduction and discussion of Dual of Truncation (DoT) that is computationally efficient and has a minimum reconstruction error.

3. The paper does a commendable job disentangling the different relevant components of SSMs (frame, measure, discretization, diagonalization, efficient implementation), and provides good explanations in general.

Weaknesses:

1. The paper is missing empirical validation demonstrating the usefulness of more general frames.

2. Section 2.3 clearly states the limitations of HiPPO, but the paper then only partially addresses these limitations.

3. Insufficient explanation how weighting and windowing measures influence the frame operator.

4. Diagonalizability of A is key to applying the method in practice. The paper has scarce discussion how choice of frame and measure influence whether A is diagonalizable.

---

> ### Author Response · Authors · 2025-09-13
> **Reponse to reviewer LAw1**
>
> Dear Reviewer,
>
> Thank you for the detailed and thoughtful feedback.  We have provided responses here as well as updates to our manuscript in blue text for ease of reference.
>
> **The paper is missing empirical validation**
>
> We have added a new section entitled “Empirical validation”. It includes instantiation of SaFARi SSMs, including orthonormal polynomial bases in prior work, plus a non-orthogonal polynomial basis, redundant frames, and functions that form neither a basis nor a frame.
>
> We present experiments comparing the performance of the SaFARi-generated SSMs with HiPPO SSMs, as well as GRUs and LSTMs.  Our goal is not a full ablation study or selection of an “optimal” frame, but to demonstrate practical implementations of our method. Table 3 and 4 are in progress; the pending results will not alter the key takeaway: a chosen frame/measure should be task-specific, and SaFARi provides that flexibility.
>
> We also include a link to our code (anonymous for the review period) that allows a user to easily construct SSMs over arbitrary frames.
>
> **The claim in Section 2.3 … is rather strong and requires more citations.**
>
> We agree this phrasing elides important differences in the task and parameterization.  We have modified this section to soften the claim and refocus on the main point:  *"Performance of various bases were shown to be strongly task-dependent. One basis-measure combination, the scaled-Legendre (LegS), performed empirically well across most tasks, but it introduced additional challenges as its $A$ matrix is not stably diagonalizable."*
>
> **Why are only uniformly weighted measures considered?**
>
> We have added the following text in Section 3.1:
>
> *"We only implement uniform weighting schemes in this work for the sake of clarity and generalizability, as the weighting does not impact the SSM's derivation.  The windowing function does impact the derivation, however, because it changes the limits of integration … one could create any arbitrary measure by combining the appropriate window with any desired weighting scheme."*
>
> **What is the use case of more general frames?**
>
> If the data exhibits patterns not aligned with classical polynomial families, information is lost.  Such bases are also ill-suited for localized phenomena, singularities, or piecewise signals.  With our method, one can easily construct an SSM from e.g. wavelets that capture localized features.
>
> **The paper has scarce discussion how choice of frame and measure influence whether $A$ is diagonalizable… this section has multiple imprecise statements… in particular the case of having degenerate eigenvalues and subspaces of eigenvectors**
>
> Our updated manuscript addresses diagonalization in Sec. 7.2. We evaluate two properties of the $A$ matrix: the sensitivity of eigendecomposition to perturbations, and the distribution (including degeneracy) of eigenvalues.  It is unknown whether it is possible to guarantee a stably diagonalizable $A$; this may depend not only on frame and measure, but also on basis quantization or ODE solver.  However, this section provides a framework for evaluation of future work, and lends support to our observations as well as that of prior work.
>
> **The explanation why A.3 has different results is not good enough.**
>
> We discovered an error in our n, k indexing.  We now find that our derivation matches that of Gu et. al. exactly, so we have removed this note.
>
> **Is Theorem 4 really a tight bound? The Frobenius norm could be quite large.**
>
> Correct; it is not a tight bound.  We have added the following text to Sec. 5.2.4:
>
> *"We note that this error bound is not necessarily tight.  Tighter bounds could be identified for specific instantiations of SSMs for given parameters (frame or basis, measure, dimension, etc.) Here, we use this generic bound to note that by keeping more elements of the frame, we can make $\epsilon$ arbitrarily small."*
>
> **How does this method benefit large-scale SSMs?**
>
> SaFARi provides flexible construction of SSMs, enabling models to capture both local and global structure, and represent long-range dependencies with complex dynamics.  These modules could be used as a drop-in replacement for the Legendre-based SSM layers in models like S4.
>
> **Does SaFARi allow for the design of new gating mechanisms in RNNs?**
>
> Gating mechanisms regulate the flow of information from past states to the current representation; SaFARi naturally lends itself to this task as online memory units that capture temporal structure. This idea aligns with recent work, such as Gu et al. (2021), which explored SSMs as gating components in long sequence models. While our paper focuses on building and analyzing SSMs, the extension to novel gating mechanisms is a promising direction for future work.
>
> **Misc.:** variables, text, and page formatting has been updated to fix the noted items.
>
> We thank the reviewer again for the meticulous attention to detail that has helped us improve the quality of our paper.

---

> > ### Comment · Reviewer_LAw1 · 2025-09-24
> > **Response to Rebuttal**
> >
> > I would like to thank the authors for their thoughtful rebuttal and updates to the manuscript. The added Section 7 "Empirical Validation" improves the submission, with a nice overview over the different frames provided in Section 7.1, and an intuitive stability characterization of the diagonalization of A in Section 7.2.
> >
> > The original questions have been satisfyingly answered in the rebuttal, but reading the new content in the manuscript led to some new questions:
> >
> > 1. In my original request I explicitly asked for "experiments from HiPPO (pMNIST etc)". Why did the authors decide to evaluate the method on a function approximation task instead?
> >
> > 2. When looking at the source code, I noticed that the authors implemented the method using numpy. Why was this library chosen? It seems that for example JAX might have been a better fit that would have allowed to scale the function to larger applications.
> >
> > 3. Table 3 shows a worse performance for the non-orthonormal frames (Bernstein, Gabor, Random Harmonics). Do the authors know about functions that would demonstrate an improved performance compared to the frames usually used (Legendre, Laguerr, Fourier, Chebyshev)?
> >
> > 4. Table 4 shows a better memorization (lower MSE) for HiPPO/Safari in SP500-delay2000 compared to SP500-delay1500. Why is that the case? I would have expected better memorization with a shorter delay.
> >
> > 5. Can the authors confirm that for Table 4 the parameter count was the same for all presented methods? GRU and LSTM seem particularly low-performing compared to the other approaches.
> >
> > Small remarks:
> >
> > 1. The manuscript mentions that "Experiments in progress to fill blanks in table as of 9/12/25.". Are the results available by now and could Tables 3,4 be completed?
> >
> > 2. The last sentence in Section 7.3.1 mentions $N_{\text{eff}}$, but this is not introduced anywhere in the text.
> >
> > 3. Page 13 - Legendre Polynomials: Legendre polynomials form an orthogonal basis, were found in prior work to perform generally well across tasks in SSMs. - not a valid english sentence
> >
> > 4. Chebyshev Polynomials: …similar structurally… -> structurally similar
> >
> > 5. All of the old bases are not only orthogonal, but also normalized, so orthonormal -> maybe the authors should change that
> >
> > 6. Figure 7 makes a comment about reproducing results from other work. Where are the related citations?
> >
> > 7. Typo in: S&P500: "price index.84"
> >
> > 8. In section 7.2 the norms need subscripts to show if they are Frobenius, L2 or whatever.

---

> ### Author Response · Authors · 2025-10-12
> **Update on experiments**
>
> Thank you very much for your additional comments.  We are providing a quick update to let you know we are in the process of re-running several of our experiments, and that you can expect a full response to your comments, along with the updated manuscript, on or before Friday, Oct. 17.
>
> The experimental results that are being updated are as follows:
>
> Table 4: The results in Table 4, as you noted, are counter-intuitive: there is a better memorization at 2000 than 1500.  We find that the MSE does increase as delay increases (as expected), but not monotonically; some initial tests indicate that our choices of 1500 and 2000 were just unlucky for this dataset, and happened to capture a short-term increase that is not reflective of the longer-term trend.  We are doing additional experiments to verify this and updating our results in the manuscript accordingly.
>
> Table 3: The results in Table 3 for the scaled version were evaluated only at the final iteration when the full signal had been observed ($t=T$, where $T$ is the length of the signal), and did not consider intermediate estimates ($t_1$, $t_2$, ... $T-1$).  While this is a valid metric, it is not directly comparable to results in the translated case, where the final iteration only considers the most recent windowed portion of the signal.  To consider the whole signal in the translated case, we must include the performance of all of the intermediate estimates, too.  In order to make the results of this table consistent, we are re-doing the scaled experiments so that intermediate reconstructions of the signal are also taken into account.  We do not expect the new results to differ significantly in magnitude from the previous, but the exact values will change.
>
> We appreciate the thorough and prompt feedback you have provided us, and look forward to providing the updated results described above, along with other corrections and responses per your review, within the week.

---

> > ### Author Response · Authors · 2025-10-17
> > **Response to additional comments (1/2)**
> >
> > Dear Reviewer,
> >
> > Thank you for your additional feedback on our revised manuscript.  As before, we have addressed your questions individually below, and posted an updated version of our paper.  For ease of reference, the text of the original manuscript is black, updates from the first revision are still in blue, and new updates are in purple.
> >
> > **1. In my original request I explicitly asked for "experiments from HiPPO (pMNIST etc)". Why did the authors decide to evaluate the method on a function approximation task instead?**
> >
> > We implemented the delay/copying task from HiPPO, based on your original comments and those of the other reviewers; we chose not to additionally pursue the pMNIST experiment.  We apologize for not addressing this specifically in our prior response, and explain our reasoning here.
> >
> > The SSM on its own is fundamentally a function approximator.  It stores a compressed history of a signal as the state vector $c$ (see Fig. 1).  The pMNIST experiment incorporated this function approximator block into an RNN with additional parameters; thus it is impossible to disambiguate the extent to which a result can be attributed to the SSM, and how much is due to the effects of learned variables or hyperparameter tuning.
> >
> > An additional complication is the relationship between an SSM’s structure and the classifier model that surrounds it. Performance depends not only on the quality of features captured by the SSM core, but also the synergy between these features and a classifier’s structure. A tree-based classifier might perform better with wavelet-based SSMs, as they capture information about localization of features, whereas a logistic-regression classifier may work better with polynomial-based SSMs.  Characterizing and understanding these complex synergies is an important question for future work, so while we felt we could not give it adequate treatment in the scope of our current paper, we do mention it in our updated conclusion.
> >
> > We also acknowledge that the original pMNIST experiment (and others) provided great value regardless of the above concerns, because they were the first to demonstrate the utility and performance of SSMs in ML contexts.  Today, SSM-based models have widespread adoption, so replicating similar results would not add the same value to our work.  In prioritizing the function approximation and delay tasks, we aim to evaluate the core functionality of the SSMs – that is, their ability to memorize long signals – independent of confounding factors.
> >
> > **2. When looking at the source code, I noticed that the authors implemented the method using numpy. Why was this library chosen? It seems that for example JAX might have been a better fit that would have allowed to scale the function to larger applications.**
> >
> > JAX is likely a better toolkit for scalability of our work, and we are considering making this change in future updates.  We have opted to use NumPy for now, because it has more widespread usage, and it defaults to higher precision on operations our method depends on (e.g. pseudo-inverse).  As our goal is to demonstrate a numerical method, and make it as portable and accessible as possible, we felt this was the appropriate choice.  However, as we move towards implementing SaFARi in large-scale machine-learning pipelines, the advantages JAX offers in GPU integration and fast gradient computation will become much more important.
> >
> > **3. Table 3 shows a worse performance for the non-orthonormal frames (Bernstein, Gabor, Random Harmonics). Do the authors know about functions that would demonstrate an improved performance compared to the frames usually used (Legendre, Laguerr, Fourier, Chebyshev)?**
> >
> > Yes! While the performance of a particular frame or basis is task-dependent, we are aware of at least one frame that outperforms the aforementioned bases in most tasks.  Our original paper had to be reduced in length and re-submitted; the material that was removed addressed this question.  We intentionally did not include that frame for two reasons:
> >
> > 1) Our goal is not to answer the question of a generically “optimal” frame or basis, but rather to provide the motivation, method, and framework to customize SSMs.
> > 2) Since the second half of the original paper is now a separate, standalone paper, we want to avoid even inadvertently violating TMLR’s ethical guidelines on originality in published material.
> >
> > *(Response continued in next comment due to character limits)*

---

> > > ### Author Response · Authors · 2025-10-17
> > > **Response to additional comments (2/2)**
> > >
> > > *(continued from previous)*
> > >
> > > **4. Table 4 shows a better memorization (lower MSE) for HiPPO/Safari in P500-delay2000 compared to SP500-delay1500. Why is that the case? I would have expected better memorization with a shorter delay.**
> > >
> > > We found that there is non-monotonic behavior of the reconstruction MSE as we vary the delay. The two data points reported in the table were correct, but did not tell the full story; while the overall MSE tends to increase with longer delays (as expected), there are local irregularities where the trend reverses.  We have replaced this table with a much more densely sampled set of delays as a plot, and included a discussion of this phenomenon in Sec 7.3.2.
> > >
> > > **5. Can the authors confirm that for Table 4 the parameter count was the same for all presented methods? GRU and LSTM seem particularly low-performing compared to the other approaches.**
> > >
> > > Yes; all models in Table 4 were configured with the same memory footprint, where the hidden or state dimension was set to $N=32$.  We have included additional discussion in Sec. 7.3.2 to address the lower performance of GRU and LSTM models.
> > >
> > > **Small remarks**
> > >
> > > **1. The manuscript mentions that "Experiments in progress to fill blanks in table as of 9/12/25.". Are the results available by now and could Tables 3,4 be completed?**
> > >
> > > These experimental results are complete; see our updated manuscript.
> > >
> > > **2. The last sentence in Section 7.3.1 mentions N_eff, but this is not introduced anywhere in the text.**
> > >
> > > Thank you for catching this; this subscript and the sentence that referenced it related to content that was removed in the previous revision.  However, given your previous question regarding performance with frames, we have opted to include a short discussion on this point (Sec 7.3.1) as well as additional experimental results, because it provides insight as to why and how redundant frames could improve performance over orthonormal bases.
> > >
> > > **3. Page 13 - Legendre Polynomials: Legendre polynomials form an orthogonal basis, were found in prior work to perform generally well across tasks in SSMs. - not a valid english sentence**
> > >
> > > Corrected.  This section is now in the appendix.
> > >
> > > **4. Chebyshev Polynomials: …similar structurally… -> structurally similar**
> > >
> > > Corrected.  This section is now in the appendix.
> > >
> > > **5. All of the old bases are not only orthogonal, but also normalized, so orthonormal -> maybe the authors should change that**
> > >
> > > Changed where appropriate.
> > >
> > > **6. Figure 7 makes a comment about reproducing results from other work. Where are the related citations?**
> > >
> > > Citations have been added here.
> > >
> > > **7. Typo in: S&P500: "price index.84"**
> > >
> > > Corrected.
> > >
> > > **8. In section 7.2 the norms need subscripts to show if they are Frobenius, L2 or whatever.**
> > >
> > > Eq. 28 is valid for any p-norm, and so we did not notate it, following the convention of the literature on the topic.  However, we agree that in this context, information about our specific choices is more useful than the generic equation.  Notation has been updated to indicate that the 2-norm was used, and a footnote clarifies that this specific norm is not a requirement of the Bauer-Fike theorem.
> > >
> > > We wish to express our gratitude again for the high level of care taken in providing this constructive feedback.  We believe you will find our updated manuscript to be improved both in content and clarity from previous submissions, and we gladly welcome any further questions or comments.

---

> > > > ### Comment · Reviewer_LAw1 · 2025-10-25
> > > >
> > > > Thank you for the detailed answers to all my questions, and the further updates to the manuscript!
> > > >
> > > > It is somewhat unfortunate that the paper had to split into two parts, with the more promising frame being completely removed from this manuscript. Since I suspect that many readers will have similar questions when reading the paper, I suggest that the authors **mention the second paper** (and the better frame) prominently with a reference, and possibly add a short section in the appendix.
> > > >
> > > > Finally, two more questions to the added content in the latest revision:
> > > >
> > > > 1. In Table 3, I find it somewhat hard to say how significant these differences really are. Would it be possible to add a rough estimation of variance, maybe by running the computation multiple times with different seeds (or on subsets of the data)?
> > > >
> > > > 2. In Figure 9, why does GRU perform better on the M4 dataset? I find the large difference surprising (and from previous difference would have bet on LSTM to be slightly better).

---

> > > > > ### Author Response · Authors · 2025-11-26
> > > > > **Response to additional comments**
> > > > >
> > > > > Dear Reviewer,
> > > > >
> > > > > Thank you for providing these additional comments.
> > > > >
> > > > > Though the manuscript has now been accepted, we wanted to take the opportunity to acknowledge your most recent feedback.
> > > > >
> > > > > **I suggest that the authors mention the second paper (and the better frame) prominently with a reference, and possibly add a short section in the appendix.**
> > > > >
> > > > > We agree that it is useful to include a reference to our other paper, as your question is a natural follow-on to the results presented in Section 7.  We have refrained from adding any significant content that would replicate work published elsewhere, but we have added a few sentences to contextualize this reference in Section 7, per your suggestion.
> > > > >
> > > > > **In Table 3, I find it somewhat hard to say how significant these differences really are. Would it be possible to add a rough estimation of variance, maybe by running the computation multiple times with different seeds (or on subsets of the data)?**
> > > > >
> > > > > Indeed, some of the values of Table 3 are not significantly different for the specific frames and data sets we chose.  Table 3 illustrates how the performance of a function approximator/memory unit depends on many factors – whether the measure is scaled or translated, the frame of choice, the rank of the SSM, and the underlying data.  For some dataset, it may well be the case that several frames or bases have equally good (or bad) performance.  But, to your point, it may also be the case that two SSMs have equally good *mean* performance on a test dataset, but one has more variance.  We have therefore included the requested standard deviation information as a table in Appendix A.9 with a short discussion.
> > > > >
> > > > > **In Figure 9, why does GRU perform better on the M4 dataset? I find the large difference surprising (and from previous difference would have bet on LSTM to be slightly better).**
> > > > >
> > > > > This is a keen observation, and the seemingly counter-intuitive result sheds some light on another advantage of SSM-based models.  GRU does outperform LSTM on the M4 dataset (See Fig. 10, right), at least for shorter delays.  While LSTMs are specifically designed to handle long sequences better than simpler models such as GRU, their added complexity makes them more sensitive to the training data.  On smaller datasets like the S&P 500, this added complexity gives negligible performance advantages.  SSM-based models, by contrast, provide a reduction in complexity of the model, and since the values of A and B are fixed, they are not sensitive to the size or content of the training data.  While we have not added additional text to the paper regarding this particular question, it is an important point that we intend to more thoroughly investigate in future works.
> > > > > We thank the reviewer again for the valuable and constructive feedback that has been provided throughout the review, and we are excited for the opportunity to share our work in TMLR.

---

### Decision · Action_Editor_W4kV · 2025-10-27

**Recommendation:** Accept as is

**Audience:**

Yes

**Audience Explanation:**

The reviewers support that this paper will be beneficial for the TMLR community and specifically for the people working on SSMs.

**Claims And Evidence:**

Yes

**Claims Explanation:**

The paper introduces SaFARi, a generalized framework for building State-Space Models (SSMs) that extends beyond the polynomial basis restrictions of current methods. The approach derives general equations for update operators as inner products, offering numerical methods for computation where closed-form solutions are not available. This work provides theoretical analysis of finite-dimensional approximation errors and demonstrates a path to computational efficiency, potentially enabling SSMs with improved stability. Despite the lack of empirical validation as raised by the reviewers, this paper is still interesting.